# Learning to Generate Projections for Reducing Dimensionality of Heterogeneous Linear Programming Problems

**Tomoharu Iwata** [1]   **Shinsaku Sakaue** [2]

## Abstract

We propose a data-driven method for reducing the dimensionality of linear programming problems (LPs) by generating instance-specific projection matrices using a neural network-based model. Once the model is trained using multiple LPs by maximizing the expected objective value, we can efficiently find high-quality feasible solutions of newly given LPs. Our method can shorten the computational time of any LP solvers due to its solver-agnostic nature, it can provide feasible solutions by relying on projection that reduces the number of variables, and it can handle LPs of different sizes using neural networks with permutation equivariance and invariance. We also provide a theoretical analysis of the generalization bound for learning a neural network to generate projection matrices that reduce the size of LPs. Our experimental results demonstrate that our method can obtain solutions with higher quality than the existing methods, while its computational time is significantly shorter than solving the original LPs.

## 1. Introduction

Linear programming (LP) is widely used in many applications, such as economics, engineering, and computer science (Gass, 2003; Eiselt & Sandblom, 2007). While numerous LP solvers have been developed, most of which are based on the simplex or interior-point methods, efficiently solving high-dimensional LPs remains an important challenge in operations research. Alongside advancements in LP solvers, there is increasing interest in reducing LP sizes through random projections (Vu et al., 2018; Poirion et al., 2023; Akchen & Mišić, 2024). Projection-based approaches are solver-agnostic in that it can work with any solvers to solve reduced-size LPs. This solver-agnostic property is particularly advantageous since we can exploit the divergent evolution of LP solvers based on the simplex and interior-point methods.

Although the random projection approach is efficient, its solution quality can be low because it is difficult for random matrices to capture the subspace of good solutions. To improve the quality, a data-driven projection approach has been proposed (Sakaue & Oki, 2024), where a projection matrix is trained using LP instances. This method produces high-quality solutions when the test LPs are sufficiently similar to the training LPs. However, it performs poorly when the test LPs differ from the training ones since it shares a projection matrix among heterogeneous LPs.

To obtain high-quality solutions efficiently for various LPs, we propose a data-driven method for learning to generate a projection matrix appropriate for each LP instance using a neural network-based model, instead of learning a projection matrix itself. Figure 1 shows our framework. In the training phase, our model is trained using multiple LPs in an end-to-end fashion. In the test phase, we are given test LPs that are related to but different from the training LPs. For each test LP instance, a projection matrix is generated using the trained model from the LP parameters. A projected LP of reduced size is obtained by the projection matrix, and it is solved by an LP solver. Then, a solution of the original test LP is recovered from the solution of the projected LP.

By generating instance-specific projected LPs based on neural networks, our method can improve the solution quality. Since our method is solver-agnostic, we can improve the efficiency of any LP solvers by reducing LP sizes. By using LP solvers that can obtain feasible solutions in the projected LPs, our method can always obtain feasible solutions since the recovered solutions are always feasible for the original LPs when the projected LPs are feasible.

Our model takes the parameters of an LP instance as input, and outputs a projection matrix. We design our model such that it is permutation equivariant to the order of the variables, and permutation invariant to the order of the constraints, which enables us to efficiently train it by reducing the model search space. Since our model can handle dif-

---

[1]NTT Corporation, Japan [2]CyberAgent, Japan. This work was done while SS was at the University of Tokyo and RIKEN AIP. Correspondence to: Tomoharu Iwata <tomoharu.iwata@ntt.com>.

*Proceedings of the 42nd International Conference on Machine Learning*, Vancouver, Canada. PMLR 267, 2025. Copyright 2025 by the author(s).

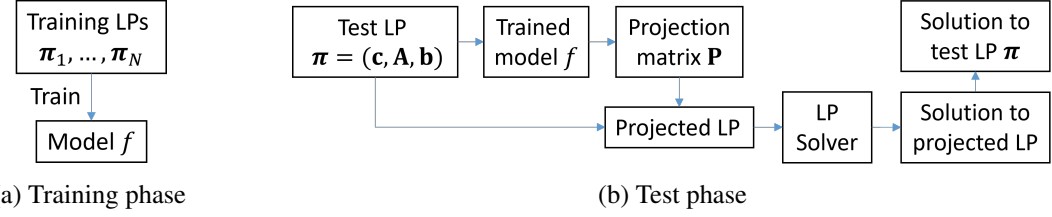

(a) Training phase             (b) Test phase

*Figure 1.* Our framework. (a) In the training phase, we train model $f$ using training LPs $\boldsymbol{\pi}_1, \ldots, \boldsymbol{\pi}_N$. (b) In the test phase, we are given a test LP $\boldsymbol{\pi}$, which is different from but related to the training LPs. Its solution is obtained via solving a reduced-size LP using instance-specific projection matrix $\mathbf{P}$ generated by trained model $f$. The size can be different across LPs.

ferent sizes of LPs, we can train it using LPs of various sizes, and use it for new LPs whose sizes are different from the training LPs. The training of our model is formulated by a bilevel optimization, where projected LPs are solved in the inner optimization, and the expected objective value of the recovered solution is maximized by updating our model's parameters in the outer optimization. We perform the bilevel optimization based on the implicit function theorem, by which we can derive an analytical expression of the outer gradients which depends only on the solution of the inner optimization and not the path taken by the inner LP solver. By synthesizing LPs according to the LP's parameter distribution of target LPs, and training our model using the synthesized LPs beforehand, we can efficiently find high-quality feasible solutions of target LPs that will be given in the future.

The main contributions of this paper are as follows: 1) We propose a framework of learning to generate instance-specific projection matrices using neural networks to efficiently solve LPs. 2) We develop a neural network model for LPs with permutation equivariance and invariance properties that can handle LPs of any sizes. 3) We provide a theoretical analysis on the generalization bound for learning a neural network to generate projection matrices that reduce the size of LPs. 4) We empirically demonstrate that our method achieves a higher quality of solutions than existing projection approaches while requiring significantly less computational time compared to solving the original LPs.

## 2. Related Work

Some existing projection-based methods for LPs reduce the number of constraints (Vu et al., 2018; Poirion et al., 2023), while others reduce the number of variables (Akchen & Mišić, 2024; Sakaue & Oki, 2024). We focus on decreasing the number of variables since the recovered solutions are always feasible for the original LPs.

Learning to optimize is an approach to accelerate optimization using machine learning techniques (Chen et al., 2023). Many methods for learning to optimize have been proposed (Monga et al., 2021; Chen et al., 2021; Amos, 2022;

Bengio et al., 2021; Nair et al., 2020; Gregor & LeCun, 2010), which use neural networks to output information useful to solvers, such as initial solutions, or to output an approximate optimal solutions for LPs (Wu & Lisser, 2023; Chen et al., 2023; Qian et al., 2024). However, the existing methods are not projection-based approaches for LPs except for Sakaue & Oki (2024). The methods directly approximate solutions by neural networks cannot guarantee the feasibility of solutions. Unlike Sakaue & Oki (2024), our method considers instance-specific projection matrices, by which we can obtain high-quality solutions of LPs of different sizes with various parameters. Although permutation equivariance in LPs has been considered using graph neural networks (Chen et al., 2023; Qian et al., 2024), they are not for generating projection matrices.

## 3. Preliminaries

We consider inequality-form LPs with parameters $\mathbf{c} \in \mathbb{R}^N$, $\mathbf{A} \in \mathbb{R}^{M \times N}$, and $\mathbf{b} \in \mathbb{R}^M$,

$$\underset{\mathbf{x} \in \mathbb{R}^N}{\text{maximize}} \quad \mathbf{c}^\top \mathbf{x} \quad \text{subject to} \quad \mathbf{A}\mathbf{x} \le \mathbf{b}, \qquad (1)$$

where $N$ is the number of variables, and $M$ is the number of constraints. An LP with equality constraints can be transformed into an inequality-form LP if a (trivially) feasible solution is available by restricting the feasible region (Sakaue & Oki, 2024)[1]. When the number of variables $N$ is large, we can efficiently compute an approximate solution of Eq. (1) by restricting variables to a low-dimensional subspace using projection matrix $\mathbf{P} \in \mathbb{R}^{N \times K}$ with $K < N$ (Poirion et al., 2023; Akchen & Mišić, 2024),

$$\underset{\mathbf{y} \in \mathbb{R}^K}{\text{maximize}} \quad \mathbf{c}^\top \mathbf{P}\mathbf{y} \quad \text{subject to} \quad \mathbf{A}\mathbf{P}\mathbf{y} \le \mathbf{b}, \qquad (2)$$

which is an LP with $K$ variables and $M$ constraints. After obtaining optimal solution $\mathbf{y}^*$ to the projected LP in Eq. (2), we can recover a solution of the original LP in Eq. (1) by

---

[1]Mathematically, we may rewrite $\mathbf{A}\mathbf{x} = \mathbf{b}$ as $\mathbf{A}\mathbf{x} \le \mathbf{b}$ and $\mathbf{A}\mathbf{x} \ge \mathbf{b}$, but this poses a challenge of identifying the subspace satisfying $\mathbf{A}\mathbf{x} = \mathbf{b}$ when generating projections. We sidestep this feasibility issue by adopting the method of Sakaue & Oki (2024).

$\tilde{\mathbf{x}} = \mathbf{P}\mathbf{y}^*$. Importantly, if the projected LP is feasible, recovered solution $\tilde{\mathbf{x}}$ is always feasible for the original LP since $\mathbf{A}\mathbf{P}\mathbf{y}^* \leq \mathbf{b}$ although it may not be optimal. The feasibility of the projected LP is guaranteed (i.e., its feasible region is non-empty) when $\mathbf{x} = \mathbf{0}$ is feasible for the original LP, since $\mathbf{y} = \mathbf{0}$ is always feasible for any projected LPs. Instead of $\mathbf{x} = \mathbf{0}$, we may assume that there exists an arbitrary common feasible solution $\mathbf{x}$ without loss of generality. This is because we can translate the feasible region so that $\mathbf{x}$ coincides with the origin $\mathbf{0}$. The quality of the solution is evaluated through objective value $\mathbf{c}^\top \mathbf{P}\mathbf{y}^*$. Ideally, if the columns of $\mathbf{P}$ span a linear subspace containing the optimal solution of (1), the reconstructed solution is optimal to (1) due to the optimality of $\mathbf{y}^*$ to (2). Thus, finding appropriate $\mathbf{P}$ that is close to this ideal condition with small $K$ enables us to efficiently compute a high-quality solution of the original LP (1) by solving the projected LP (2).

# 4. Proposed Method

We propose a neural network-based model to generate instance-specific projection matrices for LPs. Once trained on multiple LPs, the model can quickly produce an appropriate projection matrix through a single forward pass.

## 4.1. Problem Formulation

In the training phase, we are given a set of LP instances $\{\boldsymbol{\pi}_d\}_{d=1}^D$, where the $d$th LP instance is represented by its parameters $\boldsymbol{\pi}_d = (\mathbf{c}_d, \mathbf{A}_d, \mathbf{b}_d)$. Here, $\mathbf{c}_d \in \mathbb{R}^{N_d}$, $\mathbf{A}_d \in \mathbb{R}^{M_d \times N_d}$, $\mathbf{b}_d \in \mathbb{R}^{M_d}$, $N_d$ is the number of variables, and $M_d$ is the number of constraints.

In the test phase, we are given a test LP instance $\boldsymbol{\pi}$ with $N$ variables and $M$ constraints, that is different from but related to the training LPs $\{\boldsymbol{\pi}_d\}_{d=1}^D$. The number of variables and the number of constraints can be different from the training LPs, $N \neq N_d$ and $M \neq M_d$. Our aim is to obtain a high-quality solution of the test LP efficiently.

## 4.2. Model

Our neural network-based model $f_{\boldsymbol{\theta}}(\boldsymbol{\pi})$ outputs projection matrix $\mathbf{P} \in \mathbb{R}^{N \times K}$ appropriate for given LP instance $\boldsymbol{\pi}$ with $N$ variables and $M$ constraints, where $\boldsymbol{\theta}$ is model parameters. Figure 2 illustrates our model, and Figure 6 in Appendix A illustrates each component of our model.

Optimal solution $\mathbf{x}^*$ of LP instance $\boldsymbol{\pi}$ is permutation equivariant to the order of variables and permutation invariant to the order of constraints; i.e., when the variables of the LP instance are permuted, the variables of the optimal solution permuted accordingly; when the constraints of the LP instance are permuted, the optimal solution is unchanged. Therefore, appropriate projection matrix $\mathbf{P}$ is permutation equivariant on variables and permutation invariant on con-

straints. To incorporate this inductive bias, we design our model $f$ such that it always ensures permutation equivariance on variables and permutation invariance on constraints. With this design, the model search space can be significantly reduced by a factor of $N!M!$, which makes training efficient.

First, our model obtains embeddings for each element of LP parameters $\mathbf{c}$, $\mathbf{A}$, and $\mathbf{b}$ using permutation equivariant linear programming problem (PELP) layers. Let $\mathbf{z}_n^{\mathrm{c}(\ell)} \in \mathbb{R}^{H^{(\ell)}}$, $\mathbf{z}_{mn}^{\mathrm{A}(\ell)} \in \mathbb{R}^{H^{(\ell)}}$, and $\mathbf{z}_m^{\mathrm{b}(\ell)} \in \mathbb{R}^{H^{(\ell)}}$ be embedding vectors at the $\ell$th layer of an element in $\mathbf{c}$, $\mathbf{A}$, and $\mathbf{b}$, respectively, where $n$ and $m$ are indices of elements. The embeddings at the zeroth layer are the LP parameters themselves, i.e., $\mathbf{z}_n^{\mathrm{c}(0)} = c_n \in \mathbb{R}$, $\mathbf{z}_{mn}^{\mathrm{A}(0)} = A_{mn} \in \mathbb{R}$, and $\mathbf{z}_m^{\mathrm{b}(0)} = b_m \in \mathbb{R}$. The PELP layer computes the embedding of the $(n, m)$th element of $\mathbf{A}$ by

$$
\mathbf{z}_{mn}^{\mathrm{A}(\ell+1)} = \sigma\Bigg( \mathbf{W}^{\mathrm{A}1(\ell)}\mathbf{z}_n^{\mathrm{c}(\ell)} + \mathbf{W}^{\mathrm{A}2(\ell)}\mathbf{z}_{mn}^{\mathrm{A}(\ell)}
$$
$$
+ \frac{\mathbf{W}^{\mathrm{A}3(\ell)}}{M}\sum_{m'=1}^M \mathbf{z}_{m'n}^{\mathrm{A}(\ell)} + \frac{\mathbf{W}^{\mathrm{A}4(\ell)}}{N}\sum_{n'=1}^N \mathbf{z}_{mn'}^{\mathrm{A}(\ell)}
$$
$$
+ \mathbf{W}^{\mathrm{A}5(\ell)}\mathbf{z}_m^{\mathrm{b}(\ell)} + \mathbf{w}^{\mathrm{A}6(\ell)} \Bigg), \tag{3}
$$

where $\mathbf{W}^{\mathrm{A}j(\ell)} \in \mathbb{R}^{H^{(\ell)} \times H^{(\ell+1)}}, j = 1, \ldots, 5$ and $\mathbf{w}^{\mathrm{A}6(\ell)} \in \mathbb{R}^{H^{(\ell+1)}}$ are parameters at the $\ell$th layer, and $\sigma$ is a nonlinear activation function. The first term transforms the embedding of the $n$th element of $\mathbf{c}$, the second term transforms the embedding of itself, the third (fourth) term transforms the embeddings of the $n$th row ($m$th column) of $\mathbf{A}$ aggregated by average pooling, the fifth term transforms the embedding of the $m$th element of $\mathbf{b}$, and the sixth term is the bias. The model parameters for this transformation are shared across all elements $n = 1, \ldots, N$, $m = 1, \ldots, M$. Similarly, the embedding of the $m$th element of $\mathbf{b}$ is computed by

$$
\mathbf{z}_m^{\mathrm{b}(\ell+1)} = \sigma\Bigg( \frac{\mathbf{W}^{\mathrm{b}1(\ell)}}{N}\sum_{n=1}^N \mathbf{z}_{mn}^{\mathrm{A}(\ell)} + \mathbf{W}^{\mathrm{b}2(\ell)}\mathbf{z}_m^{\mathrm{b}(\ell)}
$$
$$
+ \frac{\mathbf{W}^{\mathrm{b}3(\ell)}}{M}\sum_{m'=1}^M \mathbf{z}_{m'}^{\mathrm{b}(\ell)} + \mathbf{w}^{\mathrm{b}4(\ell)} \Bigg), \tag{4}
$$

where $\mathbf{W}^{\mathrm{b}j(\ell)} \in \mathbb{R}^{H^{(\ell)} \times H^{(\ell+1)}}, j = 1, \ldots, 3$, and $\mathbf{w}^{\mathrm{b}4(\ell)} \in \mathbb{R}^{H^{(\ell+1)}}$. The embedding of the $n$th element of $\mathbf{c}$ is computed by

$$
\mathbf{z}_n^{\mathrm{c}(\ell+1)} = \sigma\Bigg( \mathbf{W}^{\mathrm{c}1(\ell)}\mathbf{z}_n^{\mathrm{c}(\ell)} + \frac{\mathbf{W}^{\mathrm{c}2(\ell)}}{N}\sum_{n'=1}^N \mathbf{z}_{n'}^{\mathrm{c}(\ell)}
$$
$$
+ \frac{\mathbf{W}^{\mathrm{c}3(\ell)}}{M}\sum_{m=1}^M \mathbf{z}_{mn}^{\mathrm{A}(\ell)} + \mathbf{w}^{\mathrm{c}4(\ell)} \Bigg), \tag{5}
$$

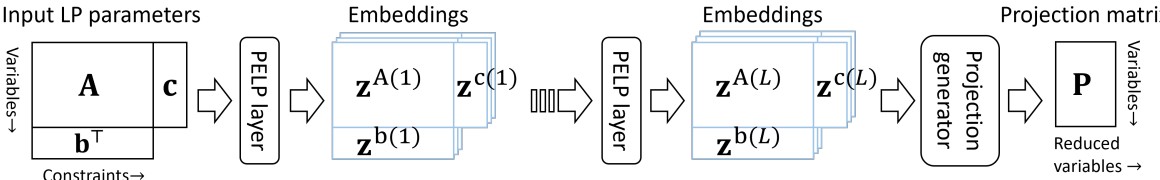

*Figure 2.* Our model takes LP parameters as input, and outputs a projection matrix.

where $\mathbf{W}^{cj(\ell)} \in \mathbb{R}^{H^{(\ell)} \times H^{(\ell+1)}}$, $j = 1, \ldots, 3$, and $\mathbf{w}^{c4(\ell)} \in \mathbb{R}^{H^{(\ell+1)}}$. Eqs. (3,4,5) are permutation equivariant on the order of variables and the order of constraints. By iterating multiple PELP layers, information in other elements is gathered through a nonlinear transformation, allowing us to obtain embeddings that incorporate information in all LP parameters, $\mathbf{c}$, $\mathbf{A}$, and $\mathbf{b}$.

After the $L$ PELP layers are applied, the $n$th row of projection matrix $\mathbf{P} = [\mathbf{p}_{1\cdot}^\top, \ldots, \mathbf{p}_{N\cdot}^\top]^\top$ is calculated by the following projection generator,

$$\mathbf{p}_{n\cdot} = g\left(\left[\mathbf{z}_n^{c(L)}, \max_m \mathbf{z}_{nm}^{A(L)}\right]\right) \in \mathbb{R}^K, \qquad (6)$$

where $g : \mathbb{R}^{2H^{(L)}} \to \mathbb{R}^K$ is a fully-connected neural network, and $\max$ is max pooling. Neural network $g$ is shared across different variables $n$. Eq. (6) is permutation equivariant on variables, and permutation invariant on constraints. Therefore, our model $f$ using PELP layers in Eqs. (3,4,5) and the projection generator in Eq. (6) is permutation equivariant on variables, and permutation invariant on constraints. Our model parameters do not depend on the number of variables $N$ or the number of constraints $M$. Therefore, it can handle LPs of different sizes; it generates a projection matrix of size $N \times K$ when an LP with $N$ variables and $M$ constraints is given for any $N$ and $M$.

Algorithm 1 shows the procedures to obtain a solution using the projection matrix generated by our model. The computational complexity of our model is $O((N + M)L)$; i.e., it linearly scales with the number of variables and constraints. Although permutation equivariance and invariance can be achieved with attention mechanisms (Lee et al., 2019), they quadratically scale with them.

### 4.3. Training

We train our model in an end-to-end fashion by maximizing the expected objective value using training LPs $\{\boldsymbol{\pi}_d\}_{d=1}^D$,

$$\hat{\boldsymbol{\theta}} = \arg\max_{\boldsymbol{\theta}} \mathbb{E}_d\left[u\left(\mathbf{P}_d, \boldsymbol{\pi}_d\right)\right], \qquad (7)$$

where $\mathbf{P}_d = f_{\boldsymbol{\theta}}(\boldsymbol{\pi}_d)$ is a projection matrix generated by our model,

$$u(\mathbf{P}, \boldsymbol{\pi}) = \max_{\mathbf{y} \in \mathbb{R}^K}\{\mathbf{c}^\top \mathbf{P} \mathbf{y} \mid \mathbf{A}\mathbf{P}\mathbf{y} \le \mathbf{b}\}, \qquad (8)$$

---

**Algorithm 1** Procedures to obtain a solution using projection matrix generated by our model.

**Input:** LP instance $\boldsymbol{\pi} = (\mathbf{c}, \mathbf{A}, \mathbf{b})$.
**Output:** Solution $\tilde{\mathbf{x}}$.
1: Initialize embeddings by the LP parameters, $\mathbf{z}_n^{c(0)} = c_n$, $\mathbf{z}_{mn}^{A(0)} = A_{mn}$, $\mathbf{z}_m^{b(0)} = b_m$ for $n = 1, \ldots, N$ and $m = 1, \ldots, M$.
2: **for** each layer $\ell = 1, \ldots, L$ **do**
3:   Compute embeddings $\{\{\mathbf{z}_{mn}^{A(\ell)}\}_{n=1}^N\}_{m=1}^M$, $\{\mathbf{z}_m^{b(\ell)}\}_{m=1}^M$, $\{\mathbf{z}_n^{c(\ell)}\}_{n=1}^N$ by the PELP layer in Eqs. (3,4,5).
4: **end for**
5: Compute projection matrix $\mathbf{P}$ by the projection generator in Eq. (6).
6: Obtain optimal solution $\mathbf{y}^*$ to projected LP $\tilde{\boldsymbol{\pi}} = (\mathbf{P}^\top \mathbf{c}, \mathbf{A}\mathbf{P}, \mathbf{b})$ by an LP solver.
7: Recover solution of original LP $\boldsymbol{\pi}$ by $\tilde{\mathbf{x}} = \mathbf{P}\mathbf{y}^*$.

---

is the optimal objective value of a projected LP, and $\mathbb{E}_d$ is the expectation over the training LPs. Model parameters $\boldsymbol{\theta}$ to be trained are the parameters in the PELP layers and the projection generator. Eq. (7) is a bilevel optimization problem, where model parameters $\boldsymbol{\theta}$ are optimized in the outer level, while reduced variables $\mathbf{y}$ of projected LPs are optimized in the inner level. To efficiently solve this bilevel optimization problem, we leverage the implicit function theorem. The gradient of the objective function in Eq. (7) for each LP instance is given by

$$\frac{\partial u(\mathbf{P}_d, \boldsymbol{\pi}_d)}{\partial \boldsymbol{\theta}} = \text{vec}\left(\frac{\partial u(\mathbf{P}_d, \boldsymbol{\pi}_d)}{\partial \mathbf{P}_d}\right)^\top \frac{\partial \text{vec}(\mathbf{P}_d)}{\partial \boldsymbol{\theta}}, \quad (9)$$

via the chain rule, where $\text{vec}: \mathbb{R}^{N \times M} \to \mathbb{R}^{NM}$ denotes the vectorization of a matrix. Assume that the projected LP satisfies a regularity condition, which requires that optimal solution $\mathbf{y}_d^* \in \mathbb{R}^K$ exists at which active constraints are linearly independent. Then, the first factor is computed based on the implicit function theorem (Tan et al., 2020; Sakaue & Oki, 2024) by

$$\frac{\partial u(\mathbf{P}_d, \boldsymbol{\pi}_d)}{\partial \mathbf{P}_d} = \mathbf{c}_d \mathbf{y}_d^{*\top} - \mathbf{A}_d^\top \boldsymbol{\lambda}_d^* \mathbf{y}_d^{*\top},$$

**Algorithm 2** Training procedures of our model.

**Input:** Training LPs $\{\boldsymbol{\pi}_d\}_{d=1}^D$.
**Output:** Trained model parameters $\boldsymbol{\theta}$.
1: **while** End condition is not satisfied **do**
2:     Sample batch of LP instances $\boldsymbol{\Pi}_{\mathrm{B}}$ from $\{\boldsymbol{\pi}_d\}$.
3:     **for** each LP instance $\boldsymbol{\pi}_d \in \boldsymbol{\Pi}_{\mathrm{B}}$ **do**
4:         Generate projection matrix $\mathbf{P}_d = f_{\boldsymbol{\theta}}(\boldsymbol{\pi}_d)$.
5:         Obtain primal $\mathbf{y}_d^*$ and dual $\boldsymbol{\lambda}_d^*$ optimal solutions of projected LP $(\mathbf{P}_d^\top \mathbf{c}_d, \mathbf{A}_d \mathbf{P}_d, \mathbf{b}_d)$ using an LP solver.
6:         Calculate gradient in Eq. (9) by automatic differentiation of $\mathrm{vec}(\mathbf{c}_d \mathbf{y}_d^{*\top} - \mathbf{A}_d^\top \boldsymbol{\lambda}_d^* \mathbf{y}_d^{*\top})^\top \mathrm{vec}(\mathbf{P}_d)$ with respect to $\boldsymbol{\theta}$.
7:     **end for**
8:     Update parameters $\boldsymbol{\theta}$ using the gradients by stochastic gradient ascent.
9: **end while**

where $\boldsymbol{\lambda}_d^* \in \mathbb{R}^M$ is the dual optimal solution of the projected LP. Here, we do not need to differentiate through the inner LP optimization path, and we can use any LP solvers that provide primal and dual optimal solutions. Eq. (9) can be computed by automatic differentiation on scalar value $\mathrm{vec}(\mathbf{c}_d \mathbf{y}_d^{*\top} - \mathbf{A}_d^\top \boldsymbol{\lambda}_d^* \mathbf{y}_d^{*\top})^\top \mathrm{vec}(\mathbf{P}_d)$ with respect to model parameters $\boldsymbol{\theta}$ without explicitly calculating large-size Jacobian $\frac{\partial \mathrm{vec}(\mathbf{P}_d)}{\partial \boldsymbol{\theta}}$. Algorithm 2 shows the training procedures of our model.

# 5. Theoretical Analysis

We discuss the generalization ability of the approach to solve LPs with projection matrices generated by neural networks based on data-driven algorithm design (Gupta & Roughgarden, 2017; Balcan, 2021; Cheng et al., 2024), which analyzes the amount of data sufficient for establishing generalization guarantees for data-driven algorithms. In particular, our analysis is based on the recent theoretical study by Cheng et al. (2024) for using neural networks in data-driven algorithm design.

## 5.1. Settings

Let $\boldsymbol{\Pi}$ be a set of LP instances with $N$ variables and $M$ constraints, where we may regard $N$ and $M$ as the largest values in $\boldsymbol{\Pi}$ to handle LPs of different sizes via zero padding. Assume that the optimal value of LPs in $\boldsymbol{\Pi}$ always lies in $[0, B]$, which is a common boundedness assumption. Let $K \in \{1, 2, \ldots, N\}$ and $\mathcal{P} = [-1, +1]^{N \times K}$ be the set of projection matrices, where the restriction to $[-1, +1]^{N \times K}$ does not lose generality since the scale of $\mathbf{P}$ does not change the optimal value of projected LPs. For any projection matrix $\mathbf{P} \in \mathcal{P}$ and instance $\boldsymbol{\pi} \in \boldsymbol{\Pi}$, define score function $u(\mathbf{P}, \boldsymbol{\pi})$ as in Eq. (8), i.e., the optimal value of the LP

instance whose feasible region is restricted to $\mathrm{Im}\,\mathbf{P}$.

For the analysis, we consider a fully-connected neural network described in Appendix B.1, leaving analysis specific to neural networks with permutation equivariance and invariance described in Section 4.2 for future work. Let $f_{\boldsymbol{\theta}}^{\mathrm{N}} : \boldsymbol{\Pi} \to \mathcal{P}$ denote the fully-connected neural network with $W$ parameters, $\boldsymbol{\theta} \in \mathbb{R}^W$, which maps an LP instance $\boldsymbol{\pi} \in \boldsymbol{\Pi}$ to an output projection matrix $\mathbf{P} \in \mathcal{P}$. Our central interest lies in the function class defined as

$$\mathcal{F}_{\mathrm{N}}^u := \left\{ u\left(f_{\boldsymbol{\theta}}^{\mathrm{N}}(\cdot), \cdot\right) : \boldsymbol{\Pi} \to [0, B] \,\big|\, \boldsymbol{\theta} \in \mathbb{R}^W \right\},$$

where each $u\left(f_{\boldsymbol{\theta}}^{\mathrm{N}}(\cdot), \cdot\right)$ is a function that is parameterized by $\boldsymbol{\theta}$ and maps $\boldsymbol{\pi} \in \boldsymbol{\Pi}$ to its optimal value under the condition that the feasible region is restricted to $\mathrm{Im}\,f_{\boldsymbol{\theta}}^{\mathrm{N}}(\boldsymbol{\pi})$.

## 5.2. Generalization Bound

We first introduce fundamental concepts in learning theory. Let $\mathcal{F}$ be a class of functions from $\boldsymbol{\Pi}$ to $\mathbb{R}$. For $D \in \mathbb{N}$, we say set $\{\boldsymbol{\pi}_d\}_{d=1}^D \subseteq \boldsymbol{\Pi}$ of input instances are *shattered* by $\mathcal{F}$ if there exist threshold values $s_1, \ldots, s_D \in \mathbb{R}$ such that

$$|\{(\mathrm{sgn}(f(\boldsymbol{\pi}_1) - s_1), \ldots, \mathrm{sgn}(f(\boldsymbol{\pi}_D) - s_D) : f \in \mathcal{F})\}| = 2^D,$$

where $\mathrm{sgn}(x) = 0$ if $x < 0$ and 1 otherwise. The *pseudo-dimension* of $\mathcal{F}$, denoted by $\mathrm{Pdim}(\mathcal{F})$, is the largest size of an input set that can be shattered by $\mathcal{F}$.

Assume that the range of functions in $\mathcal{F}$ is restricted to $[0, B]$. The celebrated uniform convergence (e.g., Anthony & Bartlett 2009, Theorem 19.2) ensures that for any distribution $\mathcal{D}$ on $\boldsymbol{\Pi}$, $\varepsilon > 0$, and $\delta \in (0, 1)$, if we have an i.i.d. sample, $\boldsymbol{\pi}_1, \ldots, \boldsymbol{\pi}_D$, from $\mathcal{D}$ of size

$$D = \Omega\left(\frac{B^2}{\varepsilon^2}\left(\mathrm{Pdim}(\mathcal{F})\log\frac{B}{\varepsilon} + \log\frac{1}{\delta}\right)\right),$$

then, with probability at least $1 - \delta$, for all $f \in \mathcal{F}$, we have

$$\left|\frac{1}{D}\sum_{d=1}^D f(\boldsymbol{\pi}_d) - \mathbb{E}_{\boldsymbol{\pi} \sim \mathcal{D}}[f(\boldsymbol{\pi})]\right| \le \varepsilon.$$

In our case, if $\mathrm{Pdim}\left(\mathcal{F}_{\mathrm{N}}^u\right)$ is bounded and the sample size is sufficiently large to satisfy $D \gtrsim \frac{B^2}{\varepsilon^2}\mathrm{Pdim}\left(\mathcal{F}_{\mathrm{N}}^u\right)$ up to log factors, then for all $\boldsymbol{\theta} \in \mathbb{R}^W$, the empirical average of $u\left(f_{\boldsymbol{\theta}}^{\mathrm{N}}(\cdot), \cdot\right)$ values deviates from the true average over unknown $\mathcal{D}$ by at most $\varepsilon$ with high probability. In other words, once we find good parameters $\boldsymbol{\theta}$ that deliver high empirical performance on a sufficiently large training set, we can expect that the trained neural network will generalize well to unseen future instances from the same distribution. Note that the bound is uniform, which holds regardless of how $\boldsymbol{\theta}$ is learned. Thanks to this uniform bound, even though the optimization of $\boldsymbol{\theta}$ is non-convex and we cannot guarantee

convergence to a globally optimal solution, once we find a model that empirically performs well on the training data, its use for unseen instances can be justified.

Let $U$ be the size of a neural network, which is the total number of hidden units. The following theorem provides an upper bound on $\mathrm{Pdim}\left(\mathcal{F}_{\mathrm{N}}^{u}\right)$, which enables us to establish the above generalization guarantee.

**Theorem 5.1.** *Let $\mathcal{P} = [-1, +1]^{N \times K}$ be a set of projection matrices, $\mathbf{\Pi}$ a set of LP instances, and $u : \mathcal{P} \times \mathbf{\Pi} \to [0, B]$ the function that takes $\mathbf{P} \in \mathcal{P}$ and $\boldsymbol{\pi} \in \mathbf{\Pi}$ as inputs and returns the optimal value of the LP instance whose feasible region is restricted to $\mathrm{Im}\,\mathbf{P}$. Let $W_0 = N + MN + M$ and $W_{L+1} = NK$, i.e., the sizes of an LP and a projection matrix, respectively. For fully-connected neural network with ReLU activations $N : \mathbb{R}^{W_0} \times \mathbb{R}^W \to \mathbb{R}^{W_{L+1}}$ with $L$ hidden layers, having size $U$ and $W$ parameters, and using the clipped ReLU squeezing function (see Appendix B.1), we have:*

$$\mathrm{Pdim}\left(\mathcal{F}_{\mathrm{N}}^{u}\right) = O(WL\log(U + NK) + WK\log(LMK)).$$

The proof is provided in Appendix B.2. This theorem allows us to estimate the amount of training data sufficient for keeping the expected test performance within an error of $\varepsilon$ via the relation of $D \gtrsim \frac{B^2}{\varepsilon^2}\mathrm{Pdim}\left(\mathcal{F}_{\mathrm{N}}^{u}\right)$, as discussed above. However, as mentioned above, the bound would have room for improvement by considering permutation equivariance and invariance of the model described in Section 4.2.

# 6. Experiments

## 6.1. Data

We evaluated our method using three types of representative LPs: packing, maximum flow, and minimum-cost flow problems, denoted by Packing, MaxFlow, and MinCostFlow, respectively. A packing problem is an LP with non-negative parameters $\mathbf{c}$, $\mathbf{A}$, and $\mathbf{b}$. We generated packing problem instances with 500 variables and 50 constraints by drawing their parameter elements from the uniform distribution on $[0, 1]$ and multiplying $\mathbf{b}$ by $N$. A maximum flow problem is an LP to find the largest flow that can be sent from a source vertex to a sink vertex in a graph while satisfying capacity constraints. A minimum-cost flow problem is an LP to find the cheapest way to send a specific amount of flow through a graph, where costs are associated with arcs. To generate an instance of MaxFlow or MinCostFlow, we first randomly created a directed graph with 50 vertices and 500 arcs with source and sink vertices, where an arc from the source to the sink was always included to ensure feasibility. For MaxFlow, the capacity of each arc was randomly drawn from the uniform distribution on $[0, 1]$. For MinCostFlow, we set the supply at the source and the demand at the sink to one. The capacity for all arcs was set to one, and the

cost for each arc was randomly drawn from the uniform distribution on $[0, 1]$. The cost between the source and sink was set to be sufficiently large. We transformed MaxFlow and MinCostFlow instances into equivalent inequality-form LPs using a trivially feasible solution (Sakaue & Oki, 2024). For MaxFlow, we used all zeros, representing no flow, as the trivially feasible solution. For MinCostFlow, we used all zeros except for a single one at the entry corresponding to the arc between the source and sink, which is trivially feasible but costly. In MaxFlow and MinCostFlow, the number of variables was 500, and the number of constraints was 1,000. All problems have non-negativity constraints $\mathbf{x} \geq 0$, which were not included in the constraint counts, and are formulated as maximization problems. For each of the Packing, MaxFlow, and MinCostFlow, we generated 425 LP instances for training, 25 for validation, and 50 for test.

## 6.2. Compared Methods

We compared our method (Ours) with the following methods: Rand, PCA, SharedP, FCNN, and Direct. Rand is a random projection matrix method (Akchen & Mišić, 2024), which reduces the dimensionality by selecting $K$ variables randomly and fixing the others to zeros. PCA is a data-driven method that obtains a projection matrix shared across LPs by applying principal component analysis to the set of optimal solutions of the training LPs (Sakaue & Oki, 2024). SharedP trains a projection matrix shared across LPs by maximizing the expected objective value on training LPs (Sakaue & Oki, 2024). FCNN generates an instance-specific projection matrix using a fully-connected neural network, where LP parameters $(\mathbf{c}, \mathbf{A}, \mathbf{b})$ are vectorized and concatenated for input. It is not permutation equivariant or invariant. Direct outputs a solution using a neural network given an LP instance, where the same model was used with our method by generating each element of a solution by the projection generator. The implementation details are described in Appendix D.1

## 6.3. Results

To evaluate the quality of the solution, we used the average objective ratio over test LP instances, where the objective ratio was the objective value of the solution divided by the optimal value of the original LP. To evaluate the efficiency, we used the computational time to obtain a solution averaged over test LP instances, which included the time required to generate a projection matrix using the trained neural network and the time taken to solve the (reduced) LP using a solver.

Figure 3 shows the average test objective ratio and computational time with different reduced dimensions $K$. Here, Full was included as a baseline that solves original $N$-dimensional LPs without reducing the dimensionality. Our

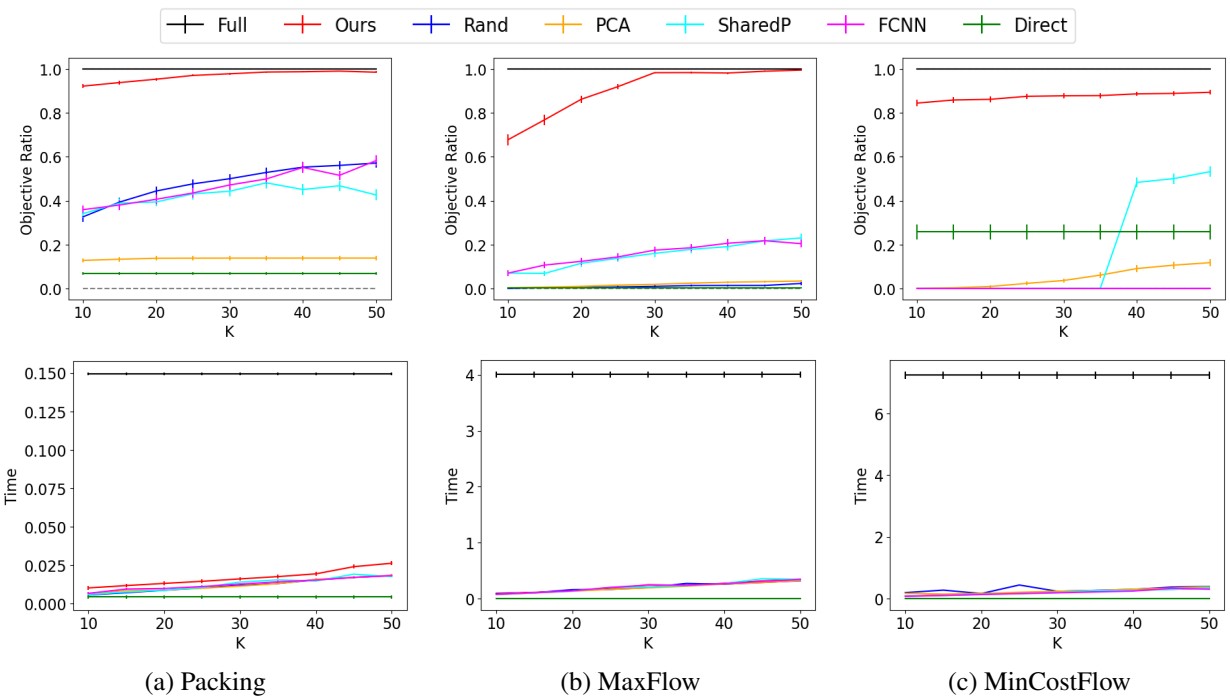

| (a) Packing | (b) MaxFlow | (c) MinCostFlow |

*Figure 3.* Average test objective ratio (upper row) and computational time in seconds for solving an LP (lower row) with different reduced dimensions $K$. Bars show the standard error.

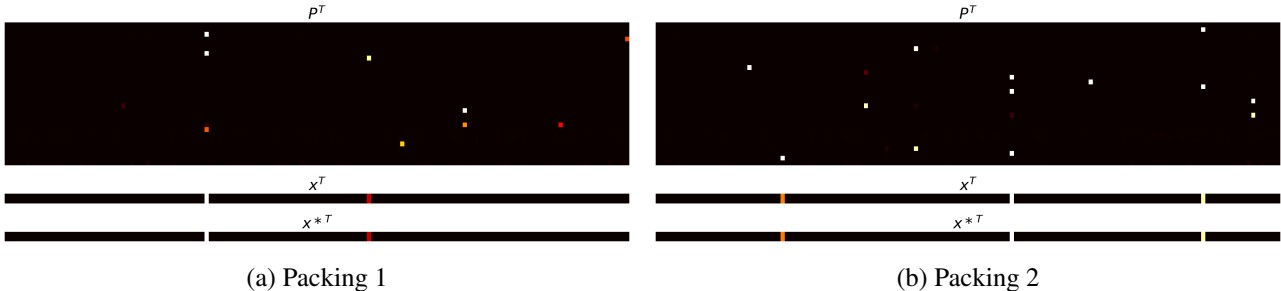

| (a) Packing 1 | (b) Packing 2 |

*Figure 4.* Examples of generated projection matrix $\mathbf{P}^\top$ using our method (top), solution $\mathbf{x}^\top$ obtained with our method (middle), and solution $\mathbf{x}^{*\top}$ obtained by solving the original LPs without dimensionality reduction (bottom) for two different packing problems. Black represents zero, which is the lowest value. The horizontal axis in all plots represents the variables, and the vertical axis in $\mathbf{P}^\top$ corresponds to the reduced variables. 150 variables out of 500 and 30 reduced variables out of 50 are shown for better visibility. The same model with $K = 50$ was used to obtain $\mathbf{P}$ and $\mathbf{x}$ in these two problems.

*Table 1.* Average test objective ratio and computational time in seconds for solving an LP on packing problems with 100,000 variables.

|  | Full | Ours | Rand | PCA | SharedP | FCNN | Direct |
|---|---|---|---|---|---|---|---|
| Objective ratio | 1.000 | 0.924 | 0.230 | 0.066 | 0.070 | 0.257 | 0.014 |
| Time | 23.487 | 0.055 | 0.013 | 0.029 | 0.063 | 0.049 | 0.005 |

*Table 2.* Average test objective ratios when types of training and test LPs were different. Mix data contain Packing, MaxFlow, and MinCostFlow.

| Test \ Train | Packing | MaxFlow | MinCostFlow | Mix |
|---|---|---|---|---|
| Packing | $0.985 \pm 0.032$ | $0.187 \pm 0.072$ | $0.164 \pm 0.084$ | $0.981 \pm 0.023$ |
| MaxFlow | $0.142 \pm 0.083$ | $0.994 \pm 0.014$ | $0.230 \pm 0.139$ | $0.954 \pm 0.058$ |
| MinCostFlow | $0.000 \pm 0.000$ | $0.029 \pm 0.083$ | $0.894 \pm 0.068$ | $0.787 \pm 0.201$ |

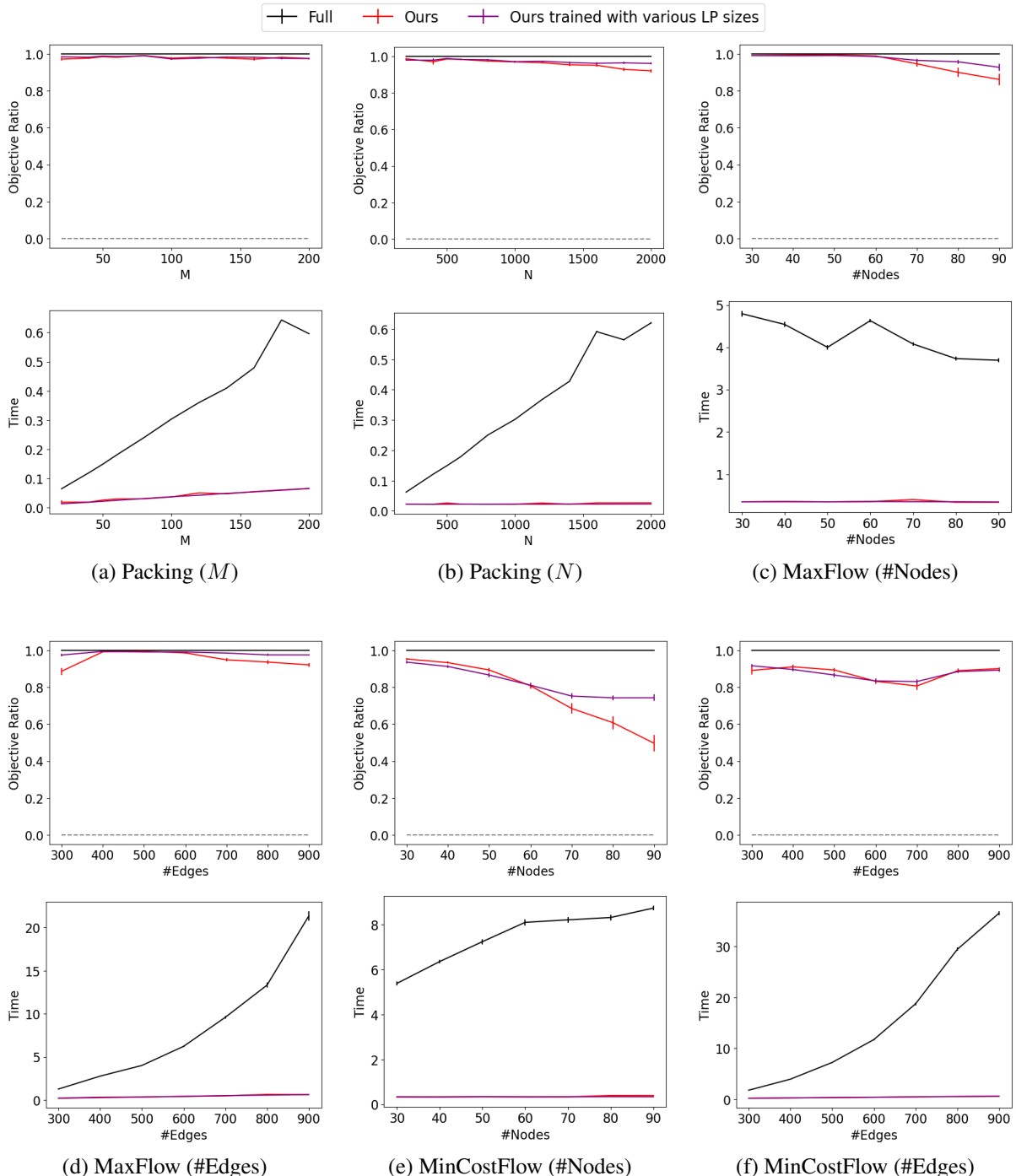

*Figure 5.* Average test objective ratio (upper row) and computational time in seconds for solving an LP (lower row) with various sizes of test LPs. Red lines (Ours) represent our method trained with LPs of a fixed size described in Section 6.1. Purple lines represent our method trained with LPs of various sizes, where the range of the training LP size was the same with the range of the test LP size. (a) The number of constraints $M$ of the test LPs was varied while fixing $N = 500$. (b) The number of variables $N$ was varied while fixing $M = 50$. (c,e) The number of nodes was varied while fixing the number of edges at 500. (d,f) The number of edges was varied while fixing the number of nodes at 50. The reduced dimension was $K = 50$.

method significantly outperformed the compared methods in terms of objective ratios. This result indicates that our

method can generate appropriate projection matrices for unseen LPs compared with the other methods. Our method

with large $K$ achieved 89% to 99% objective ratio and was much faster than Full. As reduced dimension $K$ increased, the objective ratio by our method improved while the computational time increased. Since Rand does not use information on the training LPs, its solution quality was low. Although PCA and SharedP are data-driven approaches, their objective ratios were low, since they use a projection matrix shared across all LPs. On the other hand, our method can generate different projection matrices tailored for input LPs. The low objective ratio of FCNN implies the effectiveness of permutation equivariance and invariance of our model. The objective ratio of Direct was lower than that of our method, which indicates that finding appropriate projection matrices is easier than directly finding optimal solutions. In addition, Direct violated constraints in 10% to 48% of the test LPs. On the other hand, our method always satisfied constraints by using LP solvers for reduced-size LPs. The computational time of projection-based methods (Ours, Rand, PCA, SharedP, and FCNN) increased as reduced dimension $K$ increased. Since Direct does not use solvers, it was faster than the other method. The training time for our method with $K = 50$ on Packing was 1.7 hours. Although it requires a long time for training, it can efficiently give high-quality solutions for unseen LPs once it is trained.

Figure 4 shows examples of projection matrices and solutions obtained by our method for two packing problems. While the same model was used in these different LPs, our method generated different projection matrices and succeeded in obtaining solutions close to the optimal solutions of the original LPs. The generated projection matrices have high values for variables that have non-zero values in the the original optimal solutions $\mathbf{x}^*$. This is necessary to obtain good solutions since we cannot recover non-zero solutions when the variables are removed by the projection.

An advantage of our method is that our model with identical parameters can generate projection matrices for LPs of different sizes. Figure 5 shows the results of our method when the test LP's sizes were varied. Here, we also evaluated our method trained with LPs of various sizes, where their ranges were the same with that of the test LP's sizes; $M \in [20, 200]$ and $N \in [200, 2000]$ in Packing, and the number of edges ranged from 30 to 90, and the number of nodes ranged from 300 to 900 in MaxFlow and MinCostFlow. First, we focus on our method trained with fixed-size LPs described in Section 6.1 (Ours). In Packing, the objective ratio by our method was high even when the number of constraints $M$ changed in the test LPs (a), while the increase of the computational time with $M$ was small compared with Full. The objective ratio was decreased gradually as the number of variables $N$ exceeded the training size $N = 500$ (b). These results indicate that our method works well for LPs when its number of variables is close to or smaller than that of the training LPs even when its number of constraints is different

in packing problems. In MaxFlow, the objective ratio was high when the number of nodes was close to or smaller than that of the training LPs (c), and the number of edges was close to that of the training LPs (d). In MinCostFlow, the objective ratio was high when the number of nodes was small (e), and it did not vary much with the number of edges (f). The increase in computational time with our method, as the number of edges grew, was substantially smaller than that with Full. Next, we focus on our method trained with varied-size LPs. In general, the performance was better than or comparable to our method trained with fixed-size LPs. This is reasonable since it was trained using LPs that were similar to test LPs.

The results on larger LPs are shown in Table 1, where packing problems with 100,000 variables and 50 constraints were used with reduced dimension $K = 50$. Our method achieved a higher objective ratio than the other methods (Rand and PCA), and significantly reduced computational time compared to solving the original LPs (Full) on these large instances.

Table 2 shows the objective ratios by our method when different types of LPs were used for training and test; e.g., our model was trained with Packing and tested with MaxFlow. We included Mix data for training, which consisted of Packing, MaxFlow, and MinCostFlow LPs. The objective ratios were high when the same type was used for training and test LPs, and they were low when the same type was not included. Although our method trained with the Mix data achieved slightly worse than that trained with the same type, it was much better than that trained with different types. It is important to include LPs in training that are similar to test LPs.

## 7. Conclusion

We proposed a method for learning to generate projection matrices for improving the efficiency of LP solvers by reducing LP sizes. Our method uses a neural network-based model that generates a projection matrix appropriate for a given LP instance. Since our model can handle any size of LPs, we can use it for newly given LPs whose sizes are different from the training LPs. Our theoretical analysis guarantees that the generalization error decreases as the training data size increases, potentially bringing it arbitrarily close to zero. Experiments demonstrate that our method can obtain high-quality solutions compared with the existing methods. For future work, we plan to combine our method with a smart predict-then-optimize framework (Elmachtoub & Grigas, 2022) by incorporating the prediction of LP parameters from contextual information.

## Acknowledgements

SS was supported by JST ERATO Grant Number JPM-JER1903.

## Impact Statement

This paper presents work whose goal is to advance the field of Machine Learning. There are many potential societal consequences of our work, none which we feel must be specifically highlighted here.

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

## A. Proposed Method

Figure 6 illustrates the PELP layer to obtain embeddings (a–c) and the projection generator (d).

Reducing the number of constraints and reducing the number of variables are equivalent from the perspective of LP duality. In our paper, we focus on reducing the number of variables purely for simplicity to avoid intricate discussions on infeasibility. However, from a duality perspective, one could also consider reducing the number of constraints.

By the LP relaxation of the given MILP, we can apply our method to MILPs. Our method cannot be directly used for reducing MILPs because we design our training procedures tailored to LPs. However, our high-level idea of learning to generate projections could be extended to solve MILP instances efficiently. Developing such extensions to MILPs is an interesting direction for future work.

## B. Theoretical Analysis

### B.1. Neural Networks

This section describes neural networks considered in the theoretical analysis. Let $\mathbb{N}$ be the set of positive integers. For any $N \in \mathbb{N}$, let $[N] = \{1, 2, \ldots, N\}$. Let $L \in \mathbb{N}$ and consider a neural network (NN) with architecture $[W_0, W_1, \ldots, W_L, W_{L+1}] \in \mathbb{N}^{L+2}$, which generates projection matrix $\mathbf{P} \in \mathcal{P}$ from LP instance $\boldsymbol{\pi} \in \boldsymbol{\Pi}$. Here, $L$ denotes the number of hidden layers, and $W_\ell$ denotes the number of hidden units of the $\ell$th layer. Let $U = W_1 + \cdots + W_L$ be the size of the NN. We encode LP instance $\boldsymbol{\pi} = (\mathbf{c}, \mathbf{A}, \mathbf{b})$ as an input vector of length $W_0 = N + MN + M$ by concatenation, denoted by $\mathrm{Enc} : \mathbb{R}^N \times \mathbb{R}^{M \times N} \times \mathbb{R}^M \to \mathbb{R}^{W_0}$. We also decode a vector of length $W_{L+1} = NK$ as a projection matrix with size $N \times K$, denoted by $\mathrm{Dec} : \mathbb{R}^{NK} \to \mathbb{R}^{N \times K}$. The NN is parameterized by $L + 1$ affine transformations $T_\ell : \mathbb{R}^{W_{\ell-1}} \to \mathbb{R}^{W_\ell}$, where $T_{L+1}$ is linear. We use $\boldsymbol{\theta} \in \mathbb{R}^W$ to denote the parameters of the NN, where $W$ is the number of the parameters. Using the NN, mapping from an LP instance to a projection matrix is given by

$$f_{\boldsymbol{\theta}}^{\mathrm{N}}(\boldsymbol{\pi}) = \mathrm{Dec}(T_{L+1}(\sigma(T_L(\cdots T_2(\sigma(T_1(\mathrm{Enc}(\boldsymbol{\pi})))) \cdots)))),$$

where $\sigma$ is the activation function. As in Cheng et al. (2024), we force the NN's output to lie in $[\eta_1, \tau_1] \times \cdots \times [\eta_{W_{L+1}}, \tau_{W_{L+1}}] \subset \mathbb{R}^{W_{L+1}}$ by applying a squeezing activation function $\sigma' : \mathbb{R} \to [0, 1]$ to each output coordinate $y_i \in \mathbb{R}$, namely, $\eta_i + (\tau_i - \eta_i)\sigma'(y_i)$ for $i = 1, \ldots, w_{L+1}$. We let $\tau_1 = \cdots = \tau_{w_{L+1}} = +1$ and $\eta_1 = \cdots = \eta_{w_{L+1}} = -1$ for simplicity. We focus on the fully-connected NNs with ReLU and clipped ReLU (CReLU) activation functions considered in Cheng et al. (2024), leaving analysis specific to permutation equivariant/invariant NNs for future work. Specifically, we use $\sigma(x) = \mathrm{ReLU}(x) = \max\{0, x\}$ and $\sigma'(x) = \mathrm{CReLU}(x) = \min\{\max\{0, x\}, 1\}$ for activation functions. As a side note, it is not difficult to replace ReLU with any piecewise linear activation function, such as the leaky ReLU, by accordingly modifying the bound on $Q$ discussed below based on Bartlett et al. (2019, Section 4).

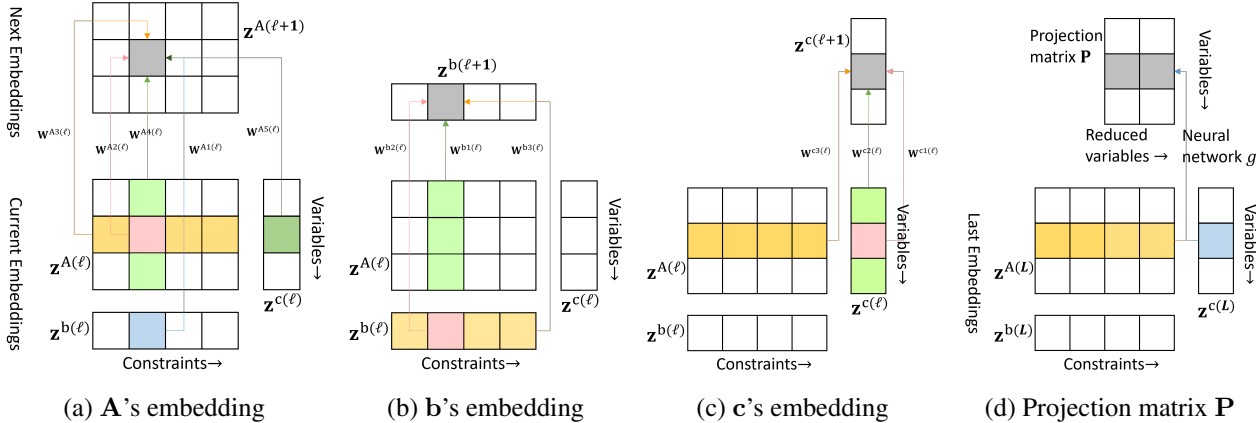

(a) $\mathbf{A}$'s embedding     (b) $\mathbf{b}$'s embedding     (c) $\mathbf{c}$'s embedding     (d) Projection matrix $\mathbf{P}$

*Figure 6.* (a–c) PELP layer to obtain embeddings of $\mathbf{A}$, $\mathbf{b}$, and $\mathbf{c}$. (d) Projection generator to obtain projection matrix $\mathbf{P}$.

## B.2. Proof

This section provides the proof of theorem 5.1.

*Proof.* As shown in the proof of Cheng et al. (2024, Theorem 2.6), for any $D \in \mathbb{N}$ and $\boldsymbol{\pi}_1, \ldots, \boldsymbol{\pi}_D \in \boldsymbol{\Pi}$, there are

$$Q \leq 2^{L+1} \left( \frac{2eD(U + 2NK)}{W} \right)^{(L+1)W}$$

subregions,$\mathcal{W}_1, \ldots, \mathcal{W}_Q$ of $\mathbb{R}^W$ such that their union is $\mathbb{R}^W$ and $f_{\boldsymbol{\theta}}^{\mathrm{N}}(\boldsymbol{\pi}_d)$ restricted to $\boldsymbol{\theta} \in \mathcal{W}_q$ is a polynomial of degree at most $L + 2$ in $\boldsymbol{\theta}$ for all $(q, d) \in [Q] \times [D]$.

Fix $(q, d) \in [Q] \times [D]$ and let $\mathbf{P}(\boldsymbol{\theta}) = f_{\boldsymbol{\theta}}^{\mathrm{N}}(\boldsymbol{\pi}_d) \in \mathcal{P}$ for any $\boldsymbol{\theta} \in \mathcal{W}_q$. Observe that $(\mathbf{P}(\boldsymbol{\theta})^{\top}\mathbf{c}, \mathbf{AP}(\boldsymbol{\theta}), \mathbf{b}) \in \mathbb{R}^K \times \mathbb{R}^{M \times K} \times \mathbb{R}^M$ is a projected LP instance such that each entry is a polynomial of degree at most $L + 2$ in $\boldsymbol{\theta}$. Then, Sakaue & Oki (2024, Lemma 4.3) implies that, for any threshold value $s \in \mathbb{R}$, there are up to $\binom{M+2K}{2K}(M + 2K + 2)$ polynomials of degree at most $(2K + 1)(L + 2)$ in $\boldsymbol{\theta}$ whose sign patterns partition $\mathcal{W}_q$ into subregions such that $u(\mathbf{P}(\boldsymbol{\theta}), \boldsymbol{\pi}_d) \geq s$ or not is identical within each subregion.

Thus, for each $q \in [Q]$ and any threshold values $s_1, \ldots, s_D \in \mathbb{R}$, by applying Warren's theorem as in the proof of Sakaue & Oki (2024, Theorem 4.4). The theorem states that, given $\gamma$ polynomials of $\nu$ variables with degrees at most $\mu$, the number of all possible sign patterns is at most $(8e\gamma\mu/\nu)^\nu$. By applying this theorem to our case, we obtain

$$\left| \left\{ \left( \mathrm{sgn}(u(f_{\boldsymbol{\theta}}^{\mathrm{N}}(\boldsymbol{\pi}_1), \boldsymbol{\pi}_1) - s_1), \ldots, \mathrm{sgn}(u(f_{\boldsymbol{\theta}}^{\mathrm{N}}(\boldsymbol{\pi}_D), \boldsymbol{\pi}_D) - s_D) \right) \ : \ \boldsymbol{\theta} \in \mathcal{W}_q \right\} \right|$$
$$\leq \left( 8eD \binom{M + 2K}{2K} \frac{(M + 2K + 2)(2K + 1)(L + 2)}{W} \right)^W .$$

Let $a_1 = 2e(U + 2NK)$ and $a_2 = 8e\binom{M+2K}{2K}(M + 2K + 2)(2K + 1)(L + 2)$ for simplicity. Summing up over $q \in [Q]$, we have

$$\left| \left\{ \left( \mathrm{sgn}(u(f_{\boldsymbol{\theta}}^{\mathrm{N}}(\boldsymbol{\pi}_1), \boldsymbol{\pi}_1) - s_1), \ldots, \mathrm{sgn}(u(f_{\boldsymbol{\theta}}^{\mathrm{N}}(\boldsymbol{\pi}_D), \boldsymbol{\pi}_D) - s_D) \right) \ : \ \boldsymbol{\theta} \in \mathbb{R}^W \right\} \right|$$
$$\leq 2^{L+1} \left( \frac{a_1 D}{W} \right)^{(L+1)W} \left( \frac{a_2 D}{W} \right)^W .$$

By definition of the pseudo-dimension, the largest possible $D$ with $2^D \leq 2^{L+1} \left( \frac{a_1 D}{W} \right)^{(L+1)W} \left( \frac{a_2 D}{W} \right)^W$ serves as an upper bound on $\mathrm{Pdim}(\mathcal{F}_{\mathrm{N}}^u)$. Taking logarithms on both sides, we obtain

$$D \log 2 \leq (L + 1) \log 2 + (L + 1)W \log \left( \frac{a_1 D}{W} \right) + W \log \left( \frac{a_2 D}{W} \right)$$
$$\leq (L + 1) \log 2 + (L + 1)W \left( \frac{D}{3(L + 1)W} + \log \left( \frac{3a_1(L + 1)}{e} \right) \right) + W \left( \frac{D}{3W} + \log \left( \frac{3a_2}{e} \right) \right)$$
$$= \frac{2}{3} D + (L + 1) \log 2 + (L + 1)W \log \left( \frac{3a_1(L + 1)}{e} \right) + W \log \left( \frac{3a_2}{e} \right),$$

where the second inequality follows from $\log z \leq \frac{z}{\lambda} + \log \frac{\lambda}{e}$ for any $z > 0$ and $\lambda > 0$ (Cheng et al., 2024, Lemma B.1). Since $\log 2 > 2/3$, the above inequality implies

$$\mathrm{Pdim}(\mathcal{F}_{\mathrm{N}}^u) \lesssim (L + 1) \log 2 + (L + 1)W \log \left( \frac{3a_1(L + 1)}{e} \right) + W \log \left( \frac{3a_2}{e} \right)$$
$$= O \left( WL \log(U + NK) + WK \log(LMK) \right),$$

completing the proof. $\qquad\square$

# C. Additional Related Work

Column generation (Desaulniers et al., 2006) is an iterative method for solving LPs: starting from an LP with a small number of variables, it iteratively selects relevant variables until the optimality is confirmed via the LP duality. Column

generation repetitively solves reduced-size LPs for solving an original LP instance. On the other hand, our method finds an appropriate projection matrix by a single forwarding pass of our neural-network-based model, and the resulting reduced-size LP is solved only once. Since column generation is an LP solver, our method can be used to accelerate column generation by benefiting from data of past LPs. Exploring the collaboration of the algorithmic (like column generation) and data-driven (like ours) approaches to reducing LP sizes will be an exciting future direction.

Mixed integer linear programming (MILP) instance generation approaches (Geng et al., 2023; Liu et al., 2024; Wang et al., 2024) generate optimization problem instances. The generated instances can be used for training machine learning-based solvers, tuning hyperparameters of solvers, or evaluating solvers. On the other hand, our method transforms a given optimization problem instance to another reduced-size instance to efficiently solve the given instance. That is, MILP instance generation approaches do not transform a given instance, while our purpose is to solve given instances efficiently. Potentially, (MI)LP instance generation techniques can be used for synthesizing training LPs.

# D. Experiments

## D.1. Implementation

In our method, we used four PELP layers with residual connection and layer normalization (Ba et al., 2016), and $H^{(\ell)} = 32$. A three-layered fully-connected neural network with 32 hidden units was used in the projection generator. The leaky rectified linear unit was used for activation functions. The softmax function is applied to the outputs of the projection generator to obtain non-negative projection matrices normalized for each reduced variable. Non-negativity constraints $\mathbf{x} \geq \mathbf{0}$ were not explicitly used for the input of our model since recovered solution $\mathbf{P}\mathbf{y}^*$ is always non-negative when projection matrix $\mathbf{P}$ is non-negative and solution $\mathbf{y}^*$ to the projected LP is non-negative. For the objective function in Eq. (8), we used the objective ratio that is the optimal objective value of the projected LP divided by the optimal value of the original LP. We optimized using Adam (Kingma & Ba, 2015) with a batch of eight LP instances, and learning rate $10^{-3}$. The maximum number of training epochs was 500, and the validation data were used for early stopping. PyTorch (Paszke et al., 2019) was used for implementation. As LP solvers, we used Gurobi 12.0.0 (Gurobi Optimization, LLC, 2024) in the training phase, and SCIP Optimization Suite 9.2.0 (Bolusani et al., 2024) in the test phase. For evaluating computational time, we used a PC with Xeon Gold 5222 3.80GHz CPU, NVIDIA GeForce RTX2080Ti GPU, and 512GB memory.

In computed projection matrices by PCA, negative values were replaced by zero to obtain non-negative projection matrices. SharedP, FCNN, and Direct were trained using the same procedures as our method. In FCNN, a three-layered fully-connected neural network with 32 hidden units was used. In Direct, the architecture of the neural network was the same as that of our method, where the projection generator was used to generate each element of a solution. In Direct, the objective function to be minimized was the mean squared error from the optimal solution $\mathbf{x}_d$ to the original LP, $\| \mathbf{x}_d^* - \mathbf{x}_d \|^2$, where $\mathbf{x}_d$ is the estimated solution. In addition, the regularization term, $|\max(\mathbf{A}_d\mathbf{x}_d - \mathbf{b}_d, 0)| + |\max(-\mathbf{x}_d, 0)|$, was added to penalize the violation of linear inequality and non-negativity constraints.

## D.2. Results

Figure 7 shows the objective ratio by our method with different numbers of training LPs. Generally, as the number of training LPs increased, the performance improved as expected. The objective ratio was high even with a relatively small number of training LPs. Additional results are provided in Appendix D.2.

Figure 8 shows the result on GROW7, an LP from the Netlib repository (Browne et al., 1995), which is a collection of software for scientific computing. We generated LP instances from GROW7 by multiplying all the LP parameters (in the objective function and constraints) with uniform random values ranging from 0.75 to 1.25 and permuting the variables and constraints. Our method achieved better objective ratios compared to the other methods, and its computational time is significantly shorter than that of solving the original LPs (Full).

Figure 9 shows the objective ratio when our method is not trained. Without training, our method did not perform well. This result demonstrates the importance of our proposed training procedures.

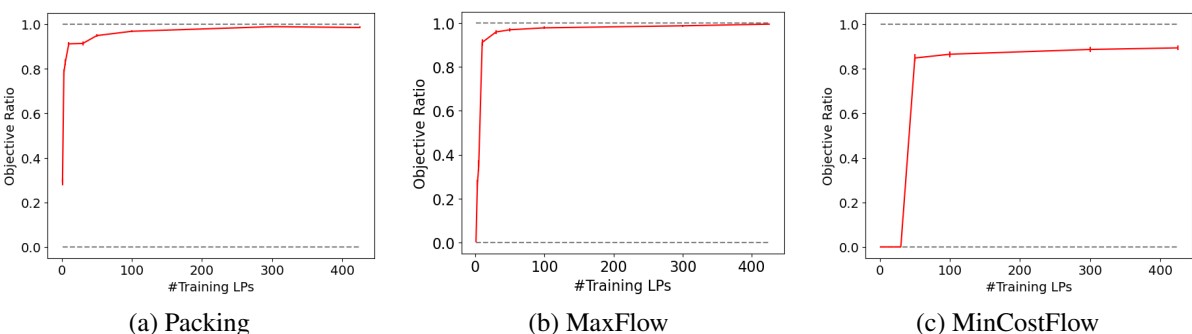

*Figure 7.* Average test objective ratio by our method with different numbers of training LPs.

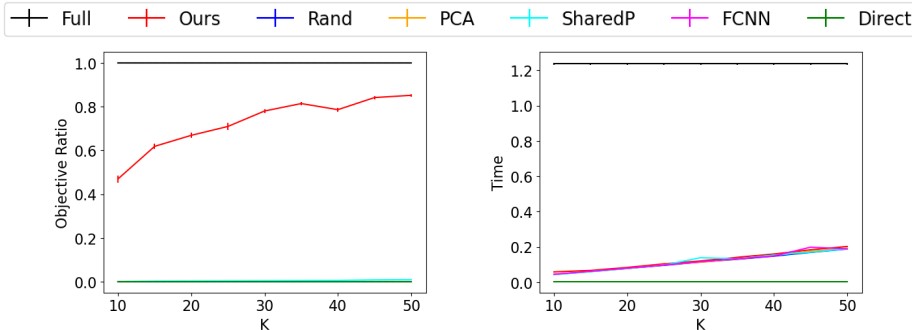

*Figure 8.* Average test objective ratio (left) and computational time in seconds for solving an LP (right) with different reduced dimension $K$ on GROW7.

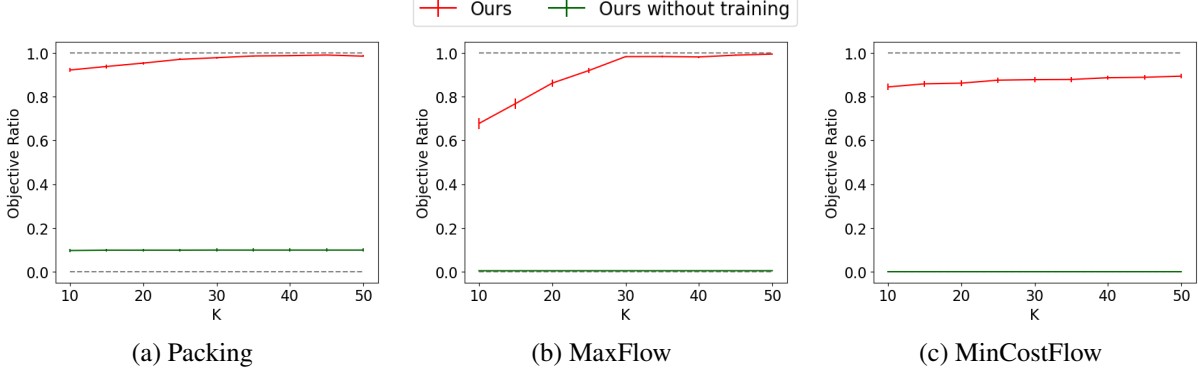

*Figure 9.* Average test objective ratio with different reduced dimension $K$ by our method with and without training.

