# OpenReview forum: "Learning to Generate Projections for Reducing Dimensionality of Heterogeneous Linear Programming Problems"
_ICML.cc/2025/Conference — ICML 2025 poster_

### Official Review · Reviewer_c6qe · 2025-03-08

**Overall Recommendation:** 3

**Summary:**

Projection is a key methodology to understand polyhedral structure and reduce dimensionality in linear programming, with recent works proposing it as a heuristic to address large-scale models. The novelty of this work is in applying a machine learning technique to learn effective projection operators. The authors present structural results associated with the sample complexity of the approach, as well as perform experiments assessing the performance against more traditional methods (e.g., PCA).

## update after rebuttal
Thank you for the rebuttal; most of my comments have been addressed. However, I am concerned with the points raised by Reviewer oitp, and prefer to slightly decrease my score as they do not seem to be addressed very clearly.

**Claims And Evidence:**

Experiments are thorough and the paper is well written and rigorously formalized.

**Essential References Not Discussed:**

N/A

**Experimental Designs Or Analyses:**

The experimental analysis is thorough. However, my minor comment is that I found the problem sizes too small (less then 2,000 variables for packing and 900 nodes for netflow). For example, state-of-the-art LP solvers can handle problems such as MaxFlow or MinCost-Flow with millions of nodes, especially as networks can be relatively sparse in many practical applications.

Although one could always argue that the time limit to solve a problem can be small, the work could better highlight the benefit of projection methods (and learning) if showcasing LPs that could not even be represented in reasonably sized memory.

**Methods And Evaluation Criteria:**

In general, every optimization approach involving a heuristic component is suitable to learning, which has fostered several ICML submissions under this same spirit. However, the differential in this paper is the nice methodology to derive an appropriate embedding for the projection matrix (Section 4.2), which I found to be non-trivial and suitable to this application. I also appreciate the paper's theoretical analysis to assess how much data is needed to derive good results.

The benchmarks were appropriately selected; in particular, they include recent work in column-randomized methods, which would be the primary methodological approach to consider within this context.

**Other Comments Or Suggestions:**

N/A

**Other Strengths And Weaknesses:**

N/A

**Questions For Authors:**

Could you please provide further interpretation of the generalization bound and training sizes that would be required for certain (carefully selected) LPs?

**Relation To Broader Scientific Literature:**

Projection methods are very classical, and their use in multiple heuristic approaches has received growing attention in these last two years. The authors fundamentally improve heuristic performance by incorporating learning.

**Theoretical Claims:**

Theorem 5.1 (the key contribution). It is based on a few seminal results in the area and I believe it is correct to the best of my analysis. However, the text tends to overly summarize much of the theoretical claims, both in the main text as well as in the proofs (e.g., when applying Warren's theorem, or discussing the many assumptions of the theorem in the main text). Although there are space issues in the main text, the proof in the Appendix could be extended to be less obscure.

My major concern is that the authors have not interpreted much of the intricate generalization bound from Section 5.2, which can be difficult to understand. For example, the text could provide a table with a (rough) estimate of the bound for some notion of large-scale LPs, and discuss how it connects to the training sets used in the experimental analysis. That is, provide further evidence of how the bound can be useful to understand the required training set sizes.

---

> ### Author Rebuttal · Authors · 2025-03-31
>
> We appreciate your positive and constructive comments.
>
> > My major concern is that the authors have not interpreted much of the intricate generalization bound from Section 5.2, which can be difficult to understand. For example, the text could provide a table with a (rough) estimate of the bound for some notion of large-scale LPs, and discuss how it connects to the training sets used in the experimental analysis. That is, provide further evidence of how the bound can be useful to understand the required training set sizes.
>
> Thank you for your suggestion. First, we must clarify that our generalization bound, as is often the case in statistical learning theory, merely provides a theoretical guarantee that the generalization error asymptotically approaches zero as the dataset size $D$ increases. We will explicitly state this point in the main text. However, with the recent rise of data-driven approaches to optimization problems, it is likely that increasingly large amounts of data will become available. Therefore, even if our bound appears impractical at present, ensuring its asymptotic convergence remains valuable.
>
> For reference, we provide the requested table of $\mathrm{Pdim}$ values corresponding to LP sizes $M, N$ and the reduced dimensionality $K$. Given an allowable generalization error $\epsilon$ and an upper bound $B$ on the LP's optimal value, an estimate for the dataset size $D$ is approximately $\frac{B^2}{\epsilon^2} Pdim$, as discussed in Section 5.2. Here, we used $L=4, K=30, N=500$, and $M=50$.
>
> |$N$|100|200|300|400|500|600|700|800|900|1000|
> |----|----|----|----|----|----|----|----|----|----|----|
> |Pdim|77039526|154435341|232369487|310654224|389199991|467953928|546881065|625956523|705161691|784482117|
>
> |$M$|100|200|300|400|500|600|700|800|900|1000|
> |----|----|----|----|----|----|----|----|----|----|----|
> |Pdim|672945915|1263418755|1874759352|2500683030|3137794899|3783981206|4437800128|5098205390|5764403103|6435770256|
>
> |$K$|10|15|20|25|30|35|40|45|50|
> |----|----|----|----|----|----|----|----|----|----|
> |Pdim|59319255|107887333|165114494|231061207|305746474|389199991|481459510|582566831|692565290|811498343|
>
> >  my minor comment is that I found the problem sizes too small (less then 2,000 variables for packing and 900 nodes for netflow).
>
> We newly conducted experiments on larger LPs (packing problems with 100,000 variables). The results for a reduced dimension of $K=50$ and number of constraints $M=50$ are shown in the table below. Our method achieved a higher objective ratio than the other methods (Rand and PCA), and significantly reduced computational time compared to solving the original LPs (Full) on these larger instances.
>
> Objective ratio
> |Ours|Rand|PCA|SharedP|FCNN|
> |----|----|----|----|----|
> |0.924|0.230|0.066|0.070|0.257|
>
> Computational time in seconds
> |Full|Ours|Rand|PCA|SharedP|FCNN|
> |----|----|----|----|----|----|
> |23.487|0.055|0.013|0.029|0.063|0.049|
>
> > Could you please provide further interpretation of the generalization bound and training sizes that would be required for certain (carefully selected) LPs?
>
> As mentioned above, if the training size $D$ is larger than $B^2/\epsilon^2 Pdim$, the generalization error is approximately bounded by $\epsilon$. Our theoretical analysis guarantees this decrease in the generalization error by increasing the training size, potentially bringing it arbitrarily close to zero.
>
> > the proof in the Appendix could be extended to be less obscure.
>
> We will extend the proof in the Appendix to make the proof more accessible to readers.

---

### Official Review · Reviewer_oitp · 2025-03-11

**Overall Recommendation:** 3

**Summary:**

This paper applies machine learning to find a more suitable projection of optimization models in the form of linear programming. By projecting to a space of smaller dimension, it is possible to solve a simpler version of the LP model much faster. A good projection should have basic solutions mapping to optimal (or near-optimal) solutions of the original LP model, in which case we may find an optimal (or near-optimal) solution from solving the project LP.

**Claims And Evidence:**

In comparison to prior work, this paper proposes a machine learning method that takes into account instance-specific information. In other words, the projection is not only based on the model but also on the specific coefficients of the LP to be projected and solve. In comparison to an approach that is not instance-specific, as well as more standard approaches, they report solutions of considerably better quality.

**Essential References Not Discussed:**

Nothing comes to mind.

**Experimental Designs Or Analyses:**

See two items above.

**Methods And Evaluation Criteria:**

I believe that evaluating the proposed NN architecture with random initialization and no training is an important benchmark. This would be different from the random projection used.

Why? Because, in recent years, some papers on ML for optimization have either spotted or reflected on the fact that ML can sometimes be an overkill. Therefore, I believe that it would a good practice that papers in this area evaluate the impact of a NN producing pure noise in their benchmarks.

**Other Comments Or Suggestions:**

Page 1:
- "is widely used in many [areas of] application" (no plural in application)
- "most of which are based": remove "are"

Page 2:
- "After obtaining [an] optimal solution"
- "if the projected LP is feasible, [the] recovered solution"

Page 3:
- "although it may not be optimal": add a comma before this excerpt
- "can be different from the training LPs, [i.e.,]"
- "Our aim is to [efficiently] obtain a high-quality solution of the test LP" (move"efficiently" from the end)
- "outputs [a] projection matrix"
- "is model parameters" -> "are the model parameters"
- "Optimal solution" -> Add "An" or "Any" in front of the sentence (LPs may have multiple optimal solutions)
- "Therefore, [an] appropriate projection"
- "which makes training [more] efficient"

Equation 6:
- Should ":" be "m"?

**Other Strengths And Weaknesses:**

Strength:
- The paper is clearly written.
- The contribution is clearly stated.

Weakness:
- Figure 3 is not very clear
- Some comments on the mathematical optimization side do not seem entirely accurate (see next items)

**Questions For Authors:**

1) When you say that "[w]e focus on decreasing the number of variables since the recovered solutions are always feasible for the original LPs", you are ignoring duality: by decreasing the number of constraints, you would find a dual feasible solution that could provide you an optimality gap. Saying that one is preferable to the other is a bit misleading. Even if not intended, I believe that addressing that by mentioning the case of decreasing the number of constraints, even if not carried out in the paper, would make it more accurate and self-contained. Remember: some people may start working on this from your paper, and such absences may affect their future work. Can you please address this?

2) What do you mean when you say that "[a]n LP with equality constraints can be transformed into an inequality-form LP if a (trivially) feasible solution is available"? Why is that even necessary? To be clear, if S = {x : A x = b}, then we can reformulate it as S = {x : A x >= b, A x <= b} = {x : - A x <= - b, A x <= b}. Isn't it?

3) You have both minimization and maximization problems, but the objective ratio always peaks at 1.0 in Figure 3. My understanding is that you take the ratio between the smallest and the largest values between the solution you obtain through the projection and the optimal solution of the LP. Is that the case? If so, can you please clarify that in the paper?

4) Please comment on what I wrote in "Methods And Evaluation Criteria".

5) Please comment on what I wrote in "Other Strengths And Weaknesses".

6) Please comment on what I wrote in "Other Comments Or Suggestions".

**Relation To Broader Scientific Literature:**

As claimed by the authors, there is a clear missing gap in using instance-specific information for obtaining a better projection for LP models.

In particular, I found the description of the PELP layers quite clear. Kudos!

**Theoretical Claims:**

Only in the appendix; not evaluated.

---

> ### Author Rebuttal · Authors · 2025-03-31
>
> We appreciate your positive and constructive comments.
>
> > I believe that evaluating the proposed NN architecture with random initialization and no training is an important benchmark. This would be different from the random projection used.
>
> We compared with the proposed NN with random initialization and no training.
> The objective ratios of our method (Ours) and our NN without training (NoTrain) are shown in the following tables. Without trainig, the proposed NN does not perform well. This result demonstrates the importance of our proposed training procedures. We will add this discussion.
>
> Packing: Objective ratio
> |$K$|10|15|20|25|30|35|40|45|50|
> |----|----|----|----|----|----|----|----|----|----|
> |Ours | 0.922 | 0.938 | 0.953 | 0.970 | 0.978 | 0.986 | 0.987 | 0.990 | 0.985 |
> |NoTrain | 0.097 | 0.098 | 0.098 | 0.098 | 0.099 | 0.099 | 0.099 | 0.099 | 0.099 |
>
> MaxFlow: Objective ratio
> |$K$|10|15|20|25|30|35|40|45|50|
> |----|----|----|----|----|----|----|----|----|----|
> |Ours | 0.677 | 0.768 | 0.862 | 0.919 | 0.983 | 0.983 | 0.982 | 0.990 | 0.994 |
> |NoTrain | 0.004 | 0.004 | 0.004 | 0.004 | 0.004 | 0.004 | 0.004 | 0.004 | 0.004 |
>
> MinCostFlow: Objective ratio
> |$K$|10|15|20|25|30|35|40|45|50|
> |----|----|----|----|----|----|----|----|----|----|
> |Ours | 0.844 | 0.859 | 0.862 | 0.875 | 0.878 | 0.879 | 0.886 | 0.889 | 0.894 |
> |NoTrain | 0.000 | 0.000 | 0.000 | 0.000 | 0.000 | 0.000 | 0.000 | 0.000 | 0.000 |
>
> > Figure 3 is not very clear
>
> We will make the figure more clearly.
>
> > Some comments on the mathematical optimization side do not seem entirely accurate
>
> We will revise the sentences according to your comments.
>
> > When you say that "[w]e focus on decreasing the number of variables since the recovered solutions are always feasible for the original LPs", you are ignoring duality: by decreasing the number of constraints, you would find a dual feasible solution that could provide you an optimality gap. Saying that one is preferable to the other is a bit misleading. Even if not intended, I believe that addressing that by mentioning the case of decreasing the number of constraints, even if not carried out in the paper, would make it more accurate and self-contained. Remember: some people may start working on this from your paper, and such absences may affect their future work. Can you please address this?
>
> Thank you for your insightful comment. As you correctly pointed out, reducing the number of constraints and reducing the number of variables are equivalent from the perspective of LP duality. In our paper, we have focused on reducing the number of variables purely for simplicity - to avoid intricate discussions on infeasibility. However, we acknowledge that, from a duality perspective, one could also consider reducing the number of constraints. We will make this explicit in the paper to ensure clarity and completeness.
>
> > What do you mean when you say that "[a]n LP with equality constraints can be transformed into an inequality-form LP if a (trivially) feasible solution is available"? Why is that even necessary? To be clear, if S = {x : A x = b}, then we can reformulate it as S = {x : A x >= b, A x <= b} = {x : - A x <= - b, A x <= b}. Isn't it?
>
> We can reformulate it as you mentioned. However, by this reformulation, projected LPs are more likely to be infeasible, and the number of inequality constraints increases. On the other hand, the reformulation using a trivially feasible solution can ease the feasibility issue, and the number of inequality constraints does not change. We will clarify this point in the paper.
>
> > You have both minimization and maximization problems, but the objective ratio always peaks at 1.0 in Figure 3. My understanding is that you take the ratio between the smallest and the largest values between the solution you obtain through the projection and the optimal solution of the LP. Is that the case? If so, can you please clarify that in the paper?
>
> In our experiments, all problems are converted into maximization problems, and $x=0$ (with an objective value of zero) is always a trivially feasible solution. Therefore, the objective ratio is calculated as the obtained objective value divided by the optimal value of the original LP. We will clarify this point in the paper.

---

> > ### Comment · Reviewer_oitp · 2025-04-06
> >
> > I do not understand your answer to one of my questions:
> >
> > _What do you mean when you say that "[a]n LP with equality constraints can be transformed into an inequality-form LP if a (trivially) feasible solution is available"? Why is that even necessary? To be clear, if S = {x : A x = b}, then we can reformulate it as S = {x : A x >= b, A x <= b} = {x : - A x <= - b, A x <= b}. Isn't it?_
> >
> > A reformulation on the same space of variables does not make the formulation more or less likely to be infeasible: the solution set remains the same. I would appreciate if you explain your reasoning for this part in more clear terms.

---

> > > ### Author Response · Authors · 2025-04-07
> > >
> > > Thank you for seeking further clarification regarding equality constraints. We would like to address your concern with a more detailed explanation.
> > >
> > > You are absolutely correct that the equality-constrained set, S = {x : Ax = b}, can be mathematically reformulated as S = {x : Ax ≤ b, Ax ≥ b}. However, in our setting of learning to generate projections, there are practical considerations that have motivated our transformation based on trivial feasible solutions:
> > >
> > > 1.	As is also discussed by Sakaue and Oki (2024), learning to generate projections that satisfy equality constraints varying across instances is challenging. Thus, following their setup, we focused on inequality-form LPs to mitigate this difficulty.
> > > 2.	Still, when a trivially feasible solution is available, we can transform an equality-constrained LP into an equivalent inequality form using the procedure described in Appendix C of Sakaue and Oki. Technically, this simply restricts the movement of variables to translations within the subspace that satisfies the equality constraints, originating from the feasible solution.
> > > 3.	The resulting inequality-constrained LP typically has a full-dimensional feasible region within the linear subspace defined by the equality constraints. This empirically reduces concerns about infeasibility when learning to generate projections.
> > > 4.	In contrast, reformulating Ax = b as Ax ≤ b and Ax ≥ b creates feasible regions that are not full-dimensional, making it more challenging to learn to generate feasible projected LPs.
> > > 5.	We adopted the approach of Sakaue and Oki due to the above empirical advantage and convenience of handling equality-constrained problems uniformly as inequality-constrained LPs.
> > >
> > > It should be noted that having full-dimensional feasible regions is not strictly necessary. In some cases, neural networks can empirically learn projections onto appropriate linear subspaces. (Hence, unlike Sakaue and Oki, the equality constraints are not necessarily identical across instances.) Additionally, learning to generate projections for equality-constrained LP instances without trivially feasible solutions remains a more challenging task.
> > >
> > > We hope this clarifies our reasoning. We will expand the discussion on this point in revision. Thank you again for prompting us to provide a more thorough explanation. Please let us know should any questions or concerns remain.

---

### Official Review · Reviewer_3Xaa · 2025-03-17

**Overall Recommendation:** 3

**Summary:**

This paper presents a data-driven method for reducing the dimensionality of linear programming (LP) problems by generating instance-specific projection matrices using a neural network-based model. The proposed approach aims to improve the efficiency of LP solvers by projecting high-dimensional LP instances into a lower-dimensional space while maintaining solution feasibility and quality. The authors conclude that their method offers an efficient and solver-agnostic approach for solving large-scale LPs, making it a practical tool for accelerating LP solvers in real-world applications.

**Claims And Evidence:**

The paper provides strong empirical and theoretical evidence for its claims, but it lacks a clear comparison with existing MILP generation methods to highlight its distinct advantages.

**Essential References Not Discussed:**

NA

**Experimental Designs Or Analyses:**

The experimental design is well-structured with diverse LP benchmarks and strong empirical validation, but it lacks direct comparisons with existing MILP generation methods to contextualize its advantages.

**Methods And Evaluation Criteria:**

The proposed methods and evaluation criteria are appropriate for LP dimensionality reduction, but the lack of direct comparison with existing MILP generation approaches limits the clarity of its broader impact.

**Other Comments Or Suggestions:**

See above.

**Other Strengths And Weaknesses:**

This paper presents a well-structured and theoretically sound approach for dimensionality reduction in linear programming (LP) problems by generating instance-specific projection matrices using a neural network. One of its key strengths is the solid theoretical foundation, particularly the generalization bound analysis, which ensures the reliability of the generated projection matrices across different LP instances. Additionally, the permutation equivariant and invariant model design enhances its flexibility, allowing it to handle LPs of varying sizes effectively. Empirical evaluations on multiple LP benchmarks demonstrate that the method achieves high-quality solutions while significantly reducing computational time compared to solving the original LPs.

However, despite these strengths, the novelty of the approach is somewhat limited. The idea of using projection-based techniques for MILP/LP problem transformation has been widely explored in the literature, and the paper does not sufficiently differentiate its method from existing MILP instance generation approaches. While the theoretical analysis is a valuable contribution, the lack of a direct comparison with other MILP generation methods makes it difficult to fully assess the impact of this work within the broader optimization community. Additionally, the practical robustness of the learned projection matrices, especially when encountering out-of-distribution LP instances, could have been further analyzed.

Overall, this paper provides a rigorous and well-executed extension of projection-based LP optimization methods, but its contribution would be stronger if it explicitly addressed its uniqueness compared to prior MILP generation methods and included more discussions on practical deployment and generalization to MILPs.

**Questions For Authors:**

See above.

**Relation To Broader Scientific Literature:**

NA

**Theoretical Claims:**

The paper provides a solid theoretical analysis, including a generalization bound for the learned projection matrices, and the proofs appear correct, though their practical implications could be further clarified.

---

> ### Author Rebuttal · Authors · 2025-03-31
>
> We appreciate your constructive comments.
>
> > it lacks a clear comparison with existing MILP generation methods to highlight its distinct advantages.
>
> Since MILP generation methods and our approach serve significantly different purposes, we cannot make direct comparison. We plan to add a discussion to clarify this point, citing relevant  references, such as [A Deep Instance Generative Framework for MILP Solvers Under Limited Data Availability, NeurIPS2023], [MILP-StuDio: MILP Instance Generation via Block Structure Decomposition, NeurIPS2024], [DIG-MILP: a Deep Instance Generator for Mixed-Integer Linear Programming with Feasibility Guarantee, TMLR, 2024]. Below is the detailed discussion.
>
> MILP instance generation approaches generate optimization problem instances. The generated instances can be used for training machine learning-based solvers, tuning hyperparameters of solvers, or evaluating solvers. On the other hand, our method transforms a given optimization problem instance to another reduced-size instance to efficiently solve the given instance. That is, MILP instance generation approaches do not transform a given instance, while our purpose is to solve a given instances efficiently, making them incomparable.
>
> > proofs appear correct, though their practical implications could be further clarified.
>
> If the training-data size $D$ is larger than $B^2/\epsilon^2 Pdim$, the generalization error is bounded by $\epsilon$. Our theoretical analysis guarantees that the generalization error decreases as the training-data size increases, potentially bringing it arbitrarily close to zero. We will clarify this practical implication.
>
> >  Additionally, the practical robustness of the learned projection matrices, especially when encountering out-of-distribution LP instances, could have been further analyzed.
>
> We have analyzed the robustness of our learned neural network-based model
> in Figure 5 (p.8) and Table 1 (p.14 in the appendix) when encountering out-of-distribution LP instances.
>
> In Figure 5, Ours (red line) represents the performance of our method on test LP instances whose sizes are different from the training LP instances. Figure 5 (a,b) show that our method works well for LPs when its number of variables is close to or smaller than that of the training LPs even when its number of constraints is different in packing problems.
> In MaxFlow, our method works well when the number of nodes was close to or smaller than that of the training LPs (c), and the number of edges was close to that of the training LPs (d). In MinCostFlow, our method works well when the number of nodes was small (e), and it did not vary much with the number of edges (f).
>
> Table 1 shows that our method's performance was high when the same type of LP instances were used for training and test LPs, and it was low when instances with the same type were not included. Although our method trained with the Mix data achieved slightly worse than that trained with the same type's instances, it was much better than that trained with different types' instances. This result indicates that it is important to include LPs in training that are similar to test LPs.
>
> > included more discussions on practical deployment
>
> On practical deployment, by synthesizing LPs according to the LP's parameter distribution of target LPs, and training our model using the synthesized LPs beforehand, we can efficiently find high-quality feasible solutions of target LPs that will be given in the future as described in Section 1.
>
> Potentially, we may be able to use (MI)LP instance generation techniques for synthesizing LPs. We will add discussions on practical deployment.
>
> > generalization to MILPs
>
> By the LP relaxation of the given MILP, we can apply our method to MILPs. Our method cannot be directly used for reducing MILPs because we design our training procedures tailored to LPs. However, our high-level idea of learning to generate projections could be extended to solve MILP instances efficiently. Developing such extensions to MILPs is an interesting direction for future work. We will add discussions on generalization to MILPs.

---

> > ### Comment · Reviewer_3Xaa · 2025-04-07
> >
> > Thanks for your reply. Your response is strong enough to persuade me. I have decided to raise my score.

---

### Official Review · Reviewer_5Q4U · 2025-03-18

**Overall Recommendation:** 3

**Summary:**

This paper proposed a new data-driven method to train a machine learning model to reduce the dimensionalities of large-scale linear programming problems (LP). This model not only can provide reduced dimension LP, by solving which one could obtain a high-quality feasible solution of the original LP, but can reduce dimensionality of LPs with different sizes. The authors also use theory to show the generalization of the model, and illustrate the performance of this model by numerical experiments.

**Claims And Evidence:**

Yes. The proof of generalization bound looks good, and the numerical experiments also support the conclusion.

**Essential References Not Discussed:**

I am not familiar with using NN to reduce dimensionality for optimization problems. However, it might be important to cite some traditional methods to reduce the dimensionality of LPs, such as column generation.

**Experimental Designs Or Analyses:**

Please see "Methods And Evaluation Criteria".

**Methods And Evaluation Criteria:**

Regarding the theoretical part, the authors only show the generalization bound but do not show the optimality of this model. However, it seems that this type of generalization bound follows many existing TCS analyses about the generalization of NNs, so this result might not be very novel. For this problem, potentially, an optimality guarantee would be more interesting.

Additionally, for the numerical experiment parts, the used cases might not be general enough. It would be better to apply this new method to some problems for the competition of commercial solvers. This mentioned comparison might show the true performance of this new method.

**Other Comments Or Suggestions:**

Please see "Methods And Evaluation Criteria" for my main concern.

Additionally, it would be great if the authors could discuss the difference between this new approach and some traditional dimensionalty reduction methods for LPs.

**Other Strengths And Weaknesses:**

Please see "Methods And Evaluation Criteria" for my main concern.

Additionally, it would be great if the authors could discuss the difference between this new approach and some traditional dimensionalty reduction methods for LPs.

**Questions For Authors:**

Please see "Methods And Evaluation Criteria" for my main concern.

Additionally, it would be great if the authors could discuss the difference between this new approach and some traditional dimensionalty reduction methods for LPs.

**Relation To Broader Scientific Literature:**

There are many different methods to reduce dimension for large-scale optimization, or linear programming, such as column generation. It seems that this paper proposed a new approach to design some novel NN layers to learn the reduced-dimension LPs, which can also provide high quality solutions.

**Theoretical Claims:**

Yes. This type of bound should be similar to some existing bounds.

---

> ### Author Rebuttal · Authors · 2025-03-31
>
> We appreciate your constructive comments.
>
> > the authors only show the generalization bound but do not show the optimality of this model.
>
> Since the objective function for training our model is non-convex, it is challenging to show the optimality of the trained model. Note that we can improve the quality of solutions by maximizing our objective function, although it may not be optimal, and our method can obtain high quality solutions as demonstrated in our experiments. The generalization bound ensures that the empirical high performance generalizes to unseen future instances.
>
> Our model is based on deep sets [Deep sets, NeurIPS2017], and deep sets are known to be universal for approximating invariant and equivariant functions [On Universal Equivariant Set Networks, ICLR2020]. Moreover, the model search space of our model is significantly reduced by a factor of N!M! by its invariant and equivariant properties as described in our paper. Therefore, although we cannot say it is the optimal model, it is a data-efficient model with high representation power for generating projections for LPs.
>
> > It would be better to apply this new method to some problems for the competition of commercial solvers.
>
> We newly conducted experiments using GROW7, an LP from the Netlib repository [The Netlib Mathematical Software Repository, D-Lib Magazine, 1995], which is a collection of software for scientific computing. We generated LP instances from GROW7 by multiplying all the LP parameters (in the objective function and constraints) with uniform random values ranging from 0.75 to 1.25 and permuting the variables and constraints. As shown in the table below, our method achieved better objective ratios compared to the other methods, and its computational time is significantly shorter than that of solving the original LPs (Full).
>
> Objective ratio
> |$K$|10|15|20|25|30|35|40|45|50|
> |----|----|----|----|----|----|----|----|----|----|
> Ours | 0.469 | 0.618 | 0.669 | 0.709 | 0.780 | 0.814 | 0.786 | 0.841 | 0.852 |
> Rand | 0.000 | 0.000 | 0.000 | 0.000 | 0.000 | 0.000 | 0.000 | 0.000 | 0.000 |
> PCA | 0.000 | 0.000 | 0.000 | 0.000 | 0.000 | 0.000 | 0.000 | 0.000 | 0.000 |
> SharedP | 0.000 | 0.002 | 0.003 | 0.004 | 0.004 | 0.005 | 0.006 | 0.008 | 0.009 |
> FCNN | 0.000 | 0.000 | 0.000 | 0.000 | 0.000 | 0.000 | 0.000 | 0.000 | 0.000 |
> Direct | 0.000 | 0.000 | 0.000 | 0.000 | 0.000 | 0.000 | 0.000 | 0.000 | 0.000 |
>
> Computational time in seconds
> |$K$|10|15|20|25|30|35|40|45|50|
> |----|----|----|----|----|----|----|----|----|----|
> Full | 1.236 | 1.236 | 1.236 | 1.236 | 1.236 | 1.236 | 1.236 | 1.236 | 1.236 |
> Ours | 0.058 | 0.067 | 0.084 | 0.104 | 0.120 | 0.141 | 0.160 | 0.184 | 0.202 |
> Rand | 0.044 | 0.060 | 0.078 | 0.096 | 0.114 | 0.130 | 0.148 | 0.169 | 0.188 |
> PCA | 0.044 | 0.062 | 0.078 | 0.099 | 0.113 | 0.132 | 0.152 | 0.176 | 0.192 |
> SharedP | 0.043 | 0.059 | 0.078 | 0.097 | 0.139 | 0.133 | 0.153 | 0.170 | 0.189 |
> FCNN | 0.045 | 0.062 | 0.080 | 0.097 | 0.116 | 0.133 | 0.148 | 0.198 | 0.189 |
> Direct | 0.004 | 0.004 | 0.004 | 0.004 | 0.004 | 0.004 | 0.004 | 0.004 | 0.004 |
>
> > I am not familiar with using NN to reduce dimensionality for optimization problems.
>
> To our knowledge, there is no existing work that uses NN to reduce dimensionality for optimization problems.
>
> > it might be important to cite some traditional methods to reduce the dimensionality of LPs, such as column generation. Additionally, it would be great if the authors could discuss the difference between this new approach and some traditional dimensionalty reduction methods for LPs.
>
> We appreciate this comment. We will cite traditional methods that reduce the dimensionality of LPs, including column generation, and add discussions on them. Below we clarify the relation to column generation.
>
> Column generation is an iterative method for solving LPs: starting from an LP with a small number of variables, it iteratively selects relevant variables until the optimality is confirmed via the LP duality. A crucial distinction is that column generation is an LP solver, whereas our model serves as a data-driven preprocessing step for reducing the dimensionality of LPs. Therefore, our method can be used to accelerate column generation by benefiting from data of past LPs. From the computational perspective, column generation repetitively solves reduced-size LPs for solving an original LP instance. On the other hand, our method finds an appropriate projection matrix by a single forwarding pass of our neural-network-based model, and the resulting reduced-size LP is solved only once. Exploring the collaboration of the algorithmic (like column generation) and data-driven (like ours) approaches to reducing LP sizes will be an exciting future direction.

---

> > ### Comment · Reviewer_5Q4U · 2025-04-05
> >
> > Thank you for the response. I am satisfied with it and would like to raise my rating.

---

### Decision · Program_Chairs · 2025-05-01

**Decision:**

Accept (poster)

**Comment:**

This paper presents a learning-based method for dimensionality reduction in LPs by generating instance-specific projection matrices. The approach is well-motivated and leverages permutation-invariant neural network architectures to enhance generalization. The paper includes a solid theoretical analysis of the generalization. All reviewers appreciate this work. That being said, the authors should fully address the theoretical concern raised by Reviewer oitp in the revision. Overall, I recommend weak acceptance.